# Unifying and Optimizing Data Values for Selection via Sequential Decision-Making

**Hongliang Chi** [1] **Qiong Wu** [2] **Zhengyi Zhou** [2] **Jonathan Light** [1] **Emily Dodwell** [2] **Yao Ma** [1]

## Abstract

Data selection has emerged as a crucial downstream application of data valuation, yet the theoretical foundations for using data values in selection remain underexplored. We reformulate data selection as a sequential decision-making problem where the optimal selection sequence arises from dynamic programming, and data values can be understood as encodings of this optimal sequence. This framework unifies and reinterprets existing methods like Data Shapley through the lens of approximate dynamic programming, revealing them as myopic linear approximations to the sequential problem. We further analyze how selection optimality degrades with utility curvature under submodularity, explaining when and why these approximations fail. To bridge theory and practice, we propose an efficient bipartite graph-based surrogate that preserves submodular structure while enabling scalable greedy selection with provable guarantees. Experiments on classical ML benchmarks and large-scale LLM fine-tuning data selection demonstrate substantial improvements over existing methods. Code is publicly available at https://github.com/frankhlchi/SeqDataVal.

## 1. Introduction

Data plays a fundamental role in modern machine learning, with recent advances heavily dependent on massive datasets (Schmidhuber, 2015; LeCun et al., 2015; Hatcher & Yu, 2018). However, not all data contributes equally to model performance, leading to the development of data valuation methods that quantify the contributions of individual training samples. The dominant approaches to data valu-

ation are grounded in cooperative game theory (Shapley, 1953; Banzhaf III, 1965; Schmeidler, 1969). In this formulation, training samples are treated as players in a cooperative game, where the utility function measures the validation performance of models trained on different data subsets. Data values are then derived by aggregating each sample's marginal contributions across various subset combinations, leading to principled methods such as Data Shapley and more (Ghorbani & Zou, 2019; Wang & Jia, 2023; Kwon & Zou, 2022).

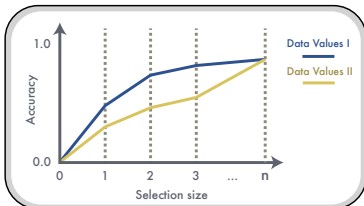

*Figure 1.* Data selection performance curves with two different data-value assignment methods.

These valuation methods provide essential guidance for crucial data-centric tasks like data selection, where the goal is to identify optimal subsets of training data that maximize model performance (Ghorbani & Zou, 2019; Schoch et al., 2023; Wang et al., 2024e). The standard protocol evaluates data values through selection curves: data points are ranked by their assigned values, and model performance is measured as more points are incrementally added in descending order of their values. As shown in Figure 1, superior data values (e.g., *Data Values I*) should show both immediate efficiency through steeper initial curves and sustained effectiveness via consistently higher performance across all selection budgets over its peer *Data Values II*.

Despite the clear importance of this selection objective, existing methods have not been explicitly optimized for this criterion. We address this gap by first establishing a fundamental optimization objective that captures the selection performance of data values across all possible subset sizes. This optimization problem can be naturally reformulated as a sequential decision-making process, where each selection decision builds upon previous choices, and the cumulative rewards directly and exactly correspond to selection performance across different subset sizes.

---

[1]Rensselaer Polytechnic Institute, Troy, NY, United States. [2]AT&T-Chief Data Office, Bedminster, NJ, United States. Correspondence to: Hongliang Chi <hc962@cornell.edu>.

*Proceedings of the 43rd International Conference on Machine Learning*, Seoul, South Korea. PMLR 306, 2026. Copyright 2026 by the author(s).

We show that this sequential perspective not only provides theoretical clarity but also naturally leads to an *exact dynamic programming (DP) solution* (Bellman, 1966) for the optimal selection sequence, from which data values can be derived as sequence encodings. Furthermore, we reveal that existing methods like Data Shapley (Ghorbani & Zou, 2019) and other semi-values (Wang & Jia, 2023; Kwon & Zou, 2022) can be understood as specific instances of *myopic linear cost (reward) function approximation* in approximate dynamic programming (ADP) (Powell, 2007; Bertsekas, 2024). Through both theoretical and empirical analysis, we show that the effectiveness of such approximations is determined by the utility function's curvature. This insight aligns with recent observations (Wang et al., 2024e) that Data Shapley performs particularly well on heterogeneous datasets. While our framework provides the optimal selection sequence through DP, its exact computation becomes intractable for large datasets. To bridge this gap between theoretical optimality and practical efficiency, we develop a novel approximation scheme based on bipartite graphs that preserves essential theoretical properties while enabling practical computation.

Specifically, our contributions are listed as follows:

- We formulate *selection curves* induced by data values as an explicit finite-horizon sequential decision problem, derive an exact DP characterization of the optimal selection sequence (from which data values arise as sequence encodings), and use this lens to reinterpret popular semi-value methods (Shapley, Banzhaf, Beta Shapley) as myopic approximate dynamic programming solutions.
- We provide new theoretical insights by analyzing existing game-theoretic data valuation methods as myopic linear approximations under our framework.
- We develop an efficient approximation scheme using bipartite surrogate utility models that preserve submodular structure, yielding provable guarantees under sufficient sandwich conditions while enabling practical computation.
- Through comprehensive experiments, we identify significant performance gaps in existing data valuation methods compared to optimal selection strategies, and demonstrate that our approximation scheme substantially closes this gap while maintaining computational efficiency.

## 2. Background and Preliminaries

### 2.1. Assumptions and Notation

We summarize here the standing assumptions and notation used throughout the paper.

**Datasets and utility functions.** We consider a finite dataset $\mathcal{D} = \{1, \ldots, n\}$ with $n = |\mathcal{D}|$. A (set) utility function is a map $U : 2^{\mathcal{D}} \to \mathbb{R}$, and we adopt the normalization $U(\emptyset) = 0$. Whenever we discuss curvature or submodularity, we additionally assume $U(\{i\}) > 0$ for all $i \in \mathcal{D}$. For a subset $S \subseteq \mathcal{D}$ and element $i \notin S$, we use the shorthand $\Delta_i U(S) \triangleq U(S \cup \{i\}) - U(S)$ for the marginal contribution of $i$ to $S$. We say $U$ is *monotone* if $U(A) \leq U(B)$ for all $A \subseteq B \subseteq \mathcal{D}$, and *submodular* if $\Delta_i U(A) \geq \Delta_i U(B)$ for all $A \subseteq B \subseteq \mathcal{D}$ and $i \in \mathcal{D} \setminus B$.

**Sequential selection and permutations.** A selection order over $\mathcal{D}$ is represented by a permutation $\pi$ of $\{1, \ldots, n\}$. We denote by $S_k^{\pi} \triangleq \{\pi(1), \ldots, \pi(k)\}$ the size-$k$ prefix induced by $\pi$. Given a value function $v : \mathcal{D} \to \mathbb{R}$, we write $\pi_v$ for the permutation that sorts points in non-increasing order of $v$, and write $S_k^v \triangleq S_k^{\pi_v}$ for the size-$k$ subset selected by ranking according to $v$. When the underlying $v$ is clear from context, we simply write $S_k$.

**Data values and marginal-contribution weights.** Game-theoretic data values assign to each point $i \in \mathcal{D}$ a score $v(i) = \sum_{S \subseteq \mathcal{D} \setminus \{i\}} \alpha_i(S) \Delta_i U(S)$, where $\alpha_i(S) \geq 0$ are *marginal-contribution weights* (which may depend on the element $i$ being valued), with normalization $\sum_S \alpha_i(S) = 1$. In semi-value based methods, these weights do not depend on $i$ and depend only on the subset size, i.e., $\alpha_i(S) = \alpha_{|S|}$, and satisfy the normalization $\sum_{k=0}^{n-1} \binom{n-1}{k} \alpha_k = 1$.

**Curvature.** For a normalized, monotone submodular function $U$ with $U(\{i\}) > 0$ for all $i$, the curvature $c \in [0, 1]$ is defined as $c \triangleq 1 - \min_{i \in \mathcal{D}} \frac{\Delta_i U(\mathcal{D} \setminus \{i\})}{\Delta_i U(\emptyset)}$. Intuitively, $c = 0$ corresponds to linear (modular) utilities, while $c = 1$ corresponds to maximal diminishing returns.

*Remark* 2.1 (Singleton-Marginal Equivalence). Under our normalization $U(\emptyset) = 0$, the singleton utility and marginal contribution at empty set coincide: $U(\{i\}) = U(\{i\}) - U(\emptyset) = \Delta_i U(\emptyset)$. This equivalence is used throughout our analysis, particularly in the curvature-related bounds of Section 4.1. Consequently, the curvature definition can be equivalently written as $c = 1 - \min_{i \in \mathcal{D}} \frac{\Delta_i U(\mathcal{D} \setminus \{i\})}{U(\{i\})}$.

### 2.2. Data Values and Data Selection

**Definition 2.2** (Score-based Data Values). Given a dataset $\mathcal{D} = \{x_1, ..., x_n\}$ and a utility function $U : 2^{\mathcal{D}} \to \mathbb{R}$ that measures the performance of any subset, the goal of data valuation is to learn a value-assignment function $v : \mathcal{D} \to \mathbb{R}$ that assigns scores to individual data points based on their contribution to the overall utility.

The predominant data values leverage solutions from the game theory, known as *Game-theoretic Data Values*. Please refer to Sections A.1 and A.3 for a broader review of other methods.

**Definition 2.3** (Game-theoretic Data Values). Given a

dataset $\mathcal{D}$ and a utility function $U$ that measures model performance on a held-out validation set when trained on different subsets, game-theoretic data values are derived by treating data points as players in a cooperative game, where for each point $x_i$, its value $v(i)$ is computed as a weighted combination of its marginal contributions across subsets:

$$v(i) = \sum_{S \subseteq \mathcal{D} \setminus \{i\}} \alpha_i(S)[U(S \cup \{i\}) - U(S)]$$

where $\alpha_i(S) \geq 0$ represents the marginal-contribution weight assigned to subset $S$ (which may depend on the element $i$ being valued), with normalization $\sum_{S \subseteq \mathcal{D} \setminus \{i\}} \alpha_i(S) = 1$.

An important special case is when the weights do not depend on $i$ and depend only on subset size, i.e., $\alpha_i(S) = \alpha_{|S|}$ for all $i$. Such values are called *semi-values* (Kwon & Zou, 2022) when they additionally satisfy $\sum_{k=0}^{n-1} \binom{n-1}{k} \alpha_k = 1$. Data Shapley is a semi-value with $\alpha_{|S|} = \frac{|S|!\,(n-|S|-1)!}{n!}$, equivalently $\alpha_k = \frac{1}{n\binom{n-1}{k}}$. Other common semi-values include:

- **Data Banzhaf**: $\alpha_k = \frac{1}{2^{n-1}}$ (uniform over all subsets)
- **Beta Shapley**: $\alpha_k^{(a,b)} = \frac{B(k+a,\,n-1-k+b)}{B(a,b)}$ for parameters $a, b > 0$

Another commonly used data value is the Leave-one-out (LOO) value $v_{\text{LOO}}(i) = U(\mathcal{D}) - U(\mathcal{D} \setminus \{i\})$, which corresponds to the degenerate size-based weighting $\alpha_{n-1} = 1$ and $\alpha_k = 0$ for $k < n - 1$. (Note that on the domain $S \subseteq \mathcal{D} \setminus \{i\}$, the unique size-$(n-1)$ subset is $\mathcal{D} \setminus \{i\}$ itself, so this size-only weighting reproduces the LOO formula.) Thus LOO fits the semi-value form, but unlike Shapley, Banzhaf, and the smooth Beta-Shapley family considered in Theorem 4.1, LOO does not arise from the population regression characterizations of that theorem; it nonetheless satisfies the conditions for Theorem 4.4.

*Remark* 2.4 (Efficiency Property). The Shapley value satisfies the *efficiency* axiom: $\sum_{i \in \mathcal{D}} v_{\text{Shap}}(i) = U(\mathcal{D}) - U(\emptyset)$. However, Banzhaf and general Beta Shapley values do *not* satisfy efficiency for arbitrary utility functions. This distinction has important implications for their regression characterizations (see Theorem 4.1).

Data values provide principled approaches for quantifying the importance of training samples, with data selection emerging as a crucial downstream application. The goal of data selection is to identify an optimal subset of training data that maximizes model performance. Recent work (Wang et al., 2024e) has formulated data selection with a fixed size constraint $S_k^* = \text{argmax}_{S \subseteq \mathcal{D}, |S|=k} U(S)$, where a subset $S$ of size $k$ is selected to maximize the utility function $U$.

The standard protocol for data selection using data values is:

**Definition 2.5** (Data Values for Data Selection). Given a dataset $\mathcal{D}$ and a utility function $U : 2^{\mathcal{D}} \to \mathbb{R}$, a value function $v : \mathcal{D} \to \mathbb{R}$ induces a *selection strategy through value ranking*: let $\pi_v$ denote the permutation that sorts samples in descending order of their values, i.e., $v(\pi_v(1)) \geq v(\pi_v(2)) \geq ... \geq v(\pi_v(n))$. The value-based selection strategy is defined as: $S_k^v = \{\pi_v(1), ..., \pi_v(k)\}$

### 2.3. Sequential Decision-Making and Markov Decision Processes

Sequential decision-making problems can be systematically modeled through Markov Decision Processes (MDPs) (Howard, 1960). An MDP is defined by a tuple $\mathcal{M} = (\mathcal{S}, \mathcal{A}, P, r)$, where $\mathcal{S}$ represents the finite state space, $\mathcal{A}$ denotes the set of possible actions, $P(s'|s, a)$ specifies the probability of transitioning to state $s'$ when taking action $a$ in state $s$, and $r : \mathcal{S} \times \mathcal{A} \to \mathbb{R}$ is the reward function.

### 2.4. Approximate Dynamic Programming

DP faces computational challenges due to the curse of dimensionality when state and action spaces grow large. Approximate Dynamic Programming (ADP) (Lee & Lee, 2004; Powell, 2007; Bertsekas, 2024) introduces approximation techniques to address these limitations. The first key strategy in ADP involves parametric function approximation to estimate both value functions and reward functions. Instead of maintaining exact values for each state-action pair $(s, a)$, ADP employs parametric approximations like $\hat{V}(s; \theta)$ for value functions and $\hat{r}(s, a; \theta)$ for finite-horizon undiscounted reward functions, where $\theta$ represents learnable parameters in linear models (Powell, 2009) or neural networks (Bertsekas & Tsitsiklis, 1996). The second strategy involves simplified decision rules at each state that consider only immediate or limited-horizon future rewards, rather than the full backward induction required by exact dynamic programming. These approximations enable ADP to handle high-dimensional MDPs by trading off exact optimality for computational tractability while maintaining solution quality through careful approximation design. For a comprehensive discussion of ADP methods, we refer readers to Appendix A.2. Among various ADP approaches to solving sequential decision-making problems, myopic reward function approximation (Powell, 2016; Rempel & Cai, 2021) represents the most computationally efficient strategy by combining both approximation strategies: it uses parameterized reward approximation $\hat{r}(s, a; \theta)$ and adopts the simplest decision rule $\sigma_{\text{myopic}} = \arg\max_{a \in \mathcal{A}} \hat{r}(s, a; \theta)$. This solution focuses solely on maximizing an approximated immediate reward for the nearest time period, ignoring how current decisions might impact future states.

## 2.5. Submodular Optimization and Optimization from Samples

Submodular optimization (Dughmi, 2009; Krause & Golovin, 2014; Bilmes, 2022) aims to maximize set functions that exhibit diminishing returns. The most closely related domain to our work, sequential submodular optimization (Asadpour et al., 2023; Tang & Yuan, 2024), focuses on optimizing over a sequence of different submodular functions, particularly targeting applications like purchase probability functions in recommendation systems. We employ submodular functions as a theoretical lens to analyze data values behavior, similar to approaches in optimization from samples literature (Balkanski et al., 2016; Balkanski & Singer, 2017; Balkanski et al., 2017a). For comprehensive reviews, we refer readers to Appendix A.4 and Appendix A.5.

## 3. Sequential Data Selection Framework

Having introduced the preliminaries, we now present our main sequential data selection framework and introduce the sequential selection problem that forms the foundation of our approach.

**Problem 3.1** (Sequential Selection Objective). Given a set $\mathcal{D}$ of data points and a utility function $U : 2^{\mathcal{D}} \to \mathbb{R}$, the sequential selection problem seeks to find an optimal permutation $\pi^*$ that maximizes the cumulative utility across all prefix sizes:

$$\pi^* = \arg\max_{\pi} \mathbb{E}_k \left[ U(S_k^\pi) \right] = \arg\max_{\pi} \frac{1}{n} \sum_{k=1}^{n} U(S_k^\pi) \quad (1)$$

where $k \sim \text{Uniform}(1, |\mathcal{D}|)$, and $\pi$ is a permutation of dataset representing the selection order.

### 3.1. Sequential Data Selection as a Deterministic Markov Decision Process

The nested constraint $S_{k-1}^\pi \subset S_k^\pi$ reveals the inherent recursive structure of Problem 3.1: each selection decision is conditioned on all previous choices, and the marginal contribution of the $k$-th selected sample depends on the composition of the existing subset. This sequential dependency naturally suggests reformulating the problem as a Deterministic Markov Decision Process (DMDP). Specifically, this is a *finite-horizon, undiscounted, deterministic* MDP with no stochastic transitions:

- State $s_t \subseteq \mathcal{D}$: the set selected so far, with $s_0 = \emptyset$.
- Action $a_t \in \mathcal{D} \setminus s_t$: the next element to add.
- Transition: $s_{t+1} = s_t \cup \{a_t\}$ (deterministic).
- Reward: $r(s_t, a_t) = U(s_t \cup \{a_t\})$.

**Remark (Reward design).** We define the per-step reward as the *prefix utility* $U(s_{t+1})$ rather than the marginal gain

$\Delta_{a_t} U(s_t)$. This choice ensures that the cumulative reward $\sum_{t=0}^{n-1} r(s_t, a_t) = \sum_{k=1}^{n} U(S_k^\pi)$ exactly equals the area-under-the-selection-curve (AUSC) objective in Problem 3.1; using marginal gains would instead yield $U(S_n^\pi) = U(\mathcal{D})$, a constant independent of $\pi$.

For any trajectory corresponding to a permutation $\pi$, the cumulative reward is $\sum_{t=0}^{n-1} r(s_t, a_t) = \sum_{t=0}^{n-1} U(s_{t+1}) = \sum_{k=1}^{n} U(S_k^\pi)$, which matches Problem 3.1 exactly. The one-step greedy (myopic) policy selects $a_t \in \arg\max_{a \in \mathcal{D} \setminus s_t} U(s_t \cup \{a\}) = \arg\max_{a \in \mathcal{D} \setminus s_t} \Delta_a U(s_t)$, where $\Delta_a U(s_t) = U(s_t \cup \{a\}) - U(s_t)$ is the marginal gain. For a linear surrogate $\hat{U}(S) = \hat{b} + \sum_{i \in S} \theta_i$, we have $\Delta_a \hat{U}(s) = \theta_a$, so the myopic policy reduces to ranking by $\theta_a$ (i.e., data values).

To solve the sequential data selection problem, which starts from an empty set and sequentially selects data points until all data is selected, we need to analyze what factors determine the selection quality. The solution quality is fundamentally determined by two key factors (see Figure 2 for a visual overview):

> (1) **Reward Modeling**: How we model and access the environment's reward signals. This could be the ground-truth utility obtained through actual model training and evaluation, or a surrogate that approximates the true utility function;
> (2) **Decision Policy**: How we make decisions based on the reward function we used. The combination of these two components determines the quality of the induced ranking, as their interactions directly impact the selection performance across different subset sizes.

Through this lens of utility modeling and decision-making, we will analyze how different solutions arise from specific choices of these two components.

### 3.2. An Exact Solution via Dynamic Programming

Under the DMDP formulation, our goal is to find an optimal policy that maximizes the cumulative reward starting from any state. Let $V(s)$ denote the optimal value function starting from state $s$, representing the maximum achievable cumulative reward from state $s$ onwards. According to the Bellman optimality equation (Bellman, 1966):

$$V(s) = \begin{cases} 0, & \text{if } |s| = n \\ \max_{a \in \mathcal{D} \setminus s} \left\{ U(s \cup \{a\}) \right. , & \text{otherwise} \\ \left. + V(s \cup \{a\}) \right\} \end{cases} \quad (2)$$

where the terminal value is $0$ since no further selections can be made, and the cumulative return is accumulated through

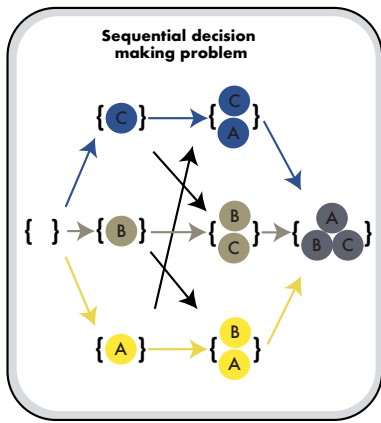
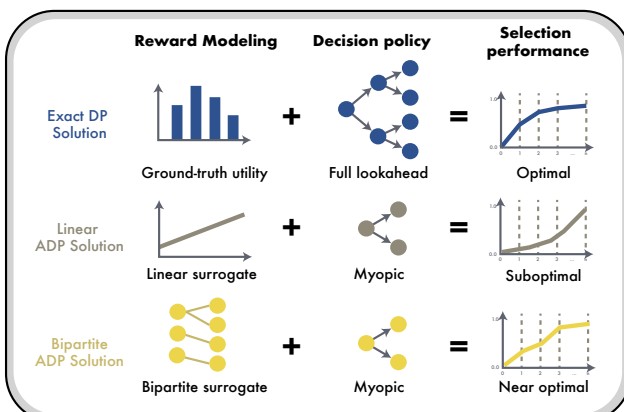

*Figure 2.* Framework for sequential data selection. Our framework consists of three components: (1) A sequential data decision problem formulating data selection through step-by-step decision-making. (2) Core components of any solution for this sequential problem including reward modeling and decision policies. (3) Selection performance curves showing outcomes from different reward modeling plus decision policy combinations, where exact DP achieves optimal performance.

the per-step rewards $U(s \cup \{a\})$.

The optimal policy $\sigma^*(s) = \text{argmax}_{a \in \mathcal{D} \setminus s}\{U(s \cup \{a\}) + V(s \cup \{a\})\}$ outputs the action that maximizes the sum of immediate reward and future value, naturally inducing an optimal selection sequence. This solution represents the ideal case where both components achieve optimality: (1) **Reward Modeling**: the ground-truth utility function $U$, and (2) **Decision Policy**: full-lookahead decision policy through immediate reward plus value function maximization as shown in Equation (2).

However, this exact solution faces significant computational challenges as it requires evaluating utilities for exponentially many subsets, motivating the need for efficient approximation methods. Specifically, the state space has $2^n$ subsets and each state has up to $n$ actions, yielding $O(n \cdot 2^n)$ state-action pairs. If each utility evaluation $U(\cdot)$ is treated as an $O(1)$ oracle call, evaluating the Bellman equation requires $O(n \cdot 2^n)$ time. In practice, however, each utility evaluation may itself be expensive (e.g., requiring model retraining), making the exact DP solution computationally prohibitive.

### 3.3. Approximated Solutions via Approximated Dynamic Programming

The exact solution through dynamic programming, while theoretically optimal, faces significant computational challenges in both key components: obtaining the ground-truth utility $U(s)$ requires model retraining for exponentially many subsets, and computing the optimal value function requires evaluating all possible future states recursively.

To address these computational challenges, we can leverage Approximate Dynamic Programming (ADP) to simplify both components. For reward modeling, instead of computing the ground-truth utility through actual model training,

we can approximate it with parametric surrogate functions. For decision making, rather than considering all future states through value function optimization, we can adopt myopic policies that only consider immediate rewards. As we will show, existing data valuation methods naturally fit into this framework through specific approximation choices.

*Remark* 3.2 (Terminology clarification). Our use of "approximate dynamic programming" here refers specifically to the combination of *approximate reward/utility modeling* (via linear surrogates) and *myopic decision policies* (one-step greedy). This differs from the classical ADP literature (Bertsekas, 2024), which typically focuses on value function approximation with multi-step lookahead or rollout policies. Our framework is more precisely described as "surrogate model + myopic policy," but we use the ADP terminology to emphasize the connection to sequential decision-making.

## 4. Analyzing Existing Data Values via ADP

Having established the optimal sequential selection framework, we now demonstrate how existing data valuation methods can be unified and analyzed through ADP. This perspective reveals that the ranking sequences induced by data values arise from specific approximation schemes in our sequential decision-making framework.

**Theorem 4.1** (Regression Views of Common Semi-Values). *Common semi-values used in data valuation coincide with regression coefficients in population regression problems. Specifically:*

- **Shapley** *values arise from constrained weighted least squares with efficiency constraint $\sum_i \theta_i = U(\mathcal{D})$ (Lundberg & Lee, 2017).*
- **Banzhaf** *values arise from unconstrained OLS under uniform subset sampling.*
- **Beta Shapley** *values arise from weighted OLS under Beta-*

*distributed sampling, for parameters $a, b > 1$ (the boundary case $a = b = 1$ reduces to Shapley and is covered by Part A).*

*The key distinction is that Shapley satisfies the efficiency axiom, while Banzhaf and Beta Shapley use unconstrained regression where an intercept absorbs the efficiency slack. Detailed formulations and proofs are provided in Appendix B and C.*

This regression perspective reveals that these methods implicitly fit a *linear surrogate* $\hat{U}(S) = \hat{b} + \sum_{i \in S} \theta_i$ to approximate the true utility function, where the coefficients $\theta_i$ correspond to data values. This insight directly connects to the ADP framework:

**Theorem 4.2** (Data Values as Myopic ADP Policies). *Let $v : \mathcal{D} \to \mathbb{R}$ be any game-theoretic value that admits a linear surrogate $\hat{U}(S) = \hat{b} + \sum_{i \in \mathcal{D}} \hat{\theta}_i \mathbb{I}_{i \in S}$ with $\hat{\theta}_i = v(i)$. Then the myopic policy $\pi_{\mathrm{myopic}}(s) \in \arg\max_{a \in \mathcal{D} \setminus s}[\hat{U}(s \cup \{a\}) - \hat{U}(s)]$ generates trajectories that sort elements in non-increasing order of $v(i)$. Equivalently, ranking by data values corresponds to myopic greedy selection under the linear surrogate. (Proof in Appendix E; scope and limitations including LOO values discussed in Appendix D.)*

### 4.1. Optimality Analysis

We now analyze when existing game-theoretic data values achieve optimal sequential selection.

**Theorem 4.3** (Optimality Under Linear Utility). *When the utility function $U(S) = \sum_{i \in S} w_i$ is linear, semi-value based methods (Data Shapley, Beta Shapley, Data Banzhaf) achieve optimal solutions to Problem 3.1. (Proof in Appendix F.)*

While linear utility represents an idealized case, real-world selection typically exhibits diminishing returns captured by submodular functions (Huh & Li, 2022). We characterize performance degradation through the curvature parameter $c \in [0, 1]$, which measures the degree of diminishing returns ($c = 0$ for linear, $c = 1$ for maximum diminishing returns).

**Theorem 4.4** (Approximation Guarantee Under Curvature). *Let $U$ be normalized, monotone, and submodular with curvature $c$. For any semi-value $v$ (Shapley, Banzhaf, Beta Shapley, LOO), let $S_k^v$ be the top-$k$ elements ranked by $v$. Then for every $k$:*

$$U(S_k^v) \geq (1 - c)^2 \, U(OPT_k^*),$$

*where $OPT_k^*$ is the unconstrained optimal size-$k$ subset. (Full definitions and proof in Appendix G.)*

The $(1 - c)^2$ bound degrades quadratically with curvature: when $c = 0$ we recover exact optimality, while as $c \to 1$ the guarantee weakens. This explains why data valuation

---

**Algorithm 1** Bipartite Graph-based Data Selection

**Require:** Training set $X_{\mathrm{train}}$, Validation set $X_{\mathrm{valid}}$, Distance function $d$
**Ensure:** Optimal threshold $\tau^*$ and ranking $\pi$
  1: Compute pairwise distances $D_{ij} = d(x_i^{\mathrm{train}}, x_j^{\mathrm{valid}})$
  2: Initialize candidate thresholds $\mathcal{T}$ from distance distribution
  3: **for** each $\tau \in \mathcal{T}$ **do**
  4:     Construct edges $E_\tau = \{(i, j) : D_{ij} \leq \tau \wedge y_i = y_j\}$
  5:     Estimate prediction error via random subset sampling
  6: **end for**
  7: Select $\tau^* = \arg\min_\tau$ prediction error
  8: Run greedy selection on $G_{\tau^*}$ to obtain ranking $\pi$
  9: **return** $\tau^*, \pi$

---

methods struggle on highly redundant datasets where points are nearly substitutable. The bound is worst-case and most useful for *explaining failures* rather than predicting success.

## 5. Efficient Approximation via Bipartite Graphs

Our analysis reveals that exact optimal selection requires exponential computation, while existing methods sacrifice optimality through linear approximations. We now develop a practical middle ground: a bipartite graph-based surrogate that preserves submodular structure while enabling efficient greedy selection with provable guarantees.

### 5.1. Bipartite Graph Construction

We construct a bipartite graph $G = (X_{\mathrm{train}}, X_{\mathrm{valid}}, E)$ between training and validation sets.

**Definition 5.1** (Coverage-based Utility). For a bipartite graph $G = (X_{\mathrm{train}}, X_{\mathrm{valid}}, E)$ with threshold $\tau$, the coverage utility of a training subset $S \subseteq X_{\mathrm{train}}$ is $\hat{U}(S) = |\{v \in X_{\mathrm{valid}} : \exists u \in S, (u, v) \in E\}|/|X_{\mathrm{valid}}|$, measuring the fraction of validation points "covered" by at least one selected training point.

**Proposition 5.2** (Submodularity of Coverage). *The coverage utility $\hat{U}(S)$ is normalized, monotone, and submodular. This guarantees that greedy selection achieves a $(1 - 1/e)$ approximation. The curvature $c \in [0, 1]$ depends on data redundancy: $c < 1$ when training points have distinct coverage, while fully redundant nodes can make $c = 1$. (Details in Appendix H.1.)*

### 5.2. Theoretical Guarantees

Coverage-based surrogates inherit the classical $(1 - 1/e)$ greedy guarantee for submodular maximization. Under idealized "sandwich" conditions where the surrogate uniformly approximates the true utility, this transfers to an end-to-

end guarantee of $(1 - \epsilon)(1 - 1/e)$ for budget-$k$ selection (Theorem H.2 in Appendix H).

In practice, these sufficient conditions may not hold exactly. Our empirical results instead support the practical claim that **the bipartite coverage surrogate is a significantly better predictor of the true utility than linear baselines**, which in turn leads to substantially improved greedy selection (see utility approximation quality in Table 3).

Algorithm 1 selects the threshold $\tau^*$ that minimizes the mean squared error between the coverage surrogate $\hat{U}(S)$ and the measured utility $U(S)$ over randomly sampled subsets $S$. While our bipartite approximation does not achieve the optimal performance of exact DP, our experiments demonstrate that it outperforms existing methods while maintaining computational efficiency.

## 6. Data Values for Selection: From Reward Functions to Value Functions

Our framework reveals a fundamental insight: data values for selection should arise from value functions rather than reward functions. Through the optimal policy derived in Section 3.2, we can define optimal data values as $v^*(i) = n - t^*(i)$, where $t^*(i)$ represents the optimal selection step for sample $i$. Unlike traditional game-theoretic data values (e.g., Shapley) that aggregate marginal contributions via fixed weighting schemes, $v^*(i)$ is an *encoding of the optimal selection sequence*: the DP solution yields an optimal permutation $\pi^*$, and we simply record each sample's position in this ordering. This value-function perspective naturally captures sequential dependencies that myopic methods miss. The encoding is not unique since any monotonic transform induces the same ranking, and when multiple optimal permutations exist, $t^*(i)$ depends on tie-breaking. While exact computation remains challenging due to exponential state space, this perspective suggests that approximate methods like lookahead policies (Bertsekas, 2024) offer promising directions beyond myopic selection.

## 7. Experiment

For evaluation, we compare our approach `Bipartite` with nine baseline data valuation methods including `Influence Function` (Koh & Liang, 2017), `Data Shapley` (Ghorbani & Zou, 2019), `Beta Shapley` (Kwon & Zou, 2022), `Data Banzhaf` (Wang & Jia, 2023), `AME` (Lin et al., 2022), `DVRL` (Yoon et al., 2020), `DataOob` (Kwon & Zou, 2023), `Leave-One-Out` (`LOO`) and `Random`, all implemented in OpenDataVal (Jiang et al., 2023) and other public dataset sources. We conduct experiments on eight diverse datasets from OpenML (Feurer et al., 2021), following the standard data valuation evaluation protocol that generates selection curves by itera-

tively adding points based on their assigned values. Unless otherwise stated, all classical OpenML experiments are averaged over 20 independent runs with 1000 model retraining steps except for the `DynamicProgramming` method (Algorithm 2), which is used for optimality verification in sequential selection. Detailed experimental settings and dataset descriptions are provided in Appendix J.

### 7.1. RQ1: How Close are Existing Data Values to Optimal Sequential Selection?

Following the experimental settings detailed in Appendix J.5, our quantitative analysis reveals substantial performance gaps between existing methods and the optimal sequential selection method `DynamicProgramming`, as summarized in Table 1. Across all datasets, existing methods consistently underperform optimal selection by 8.39% to 23.76%. The largest gap appears in structured datasets, particularly bbc-embeddings where the best baseline falls short by 23.76%. For detailed analysis of selection curves and method-specific behaviors across different data budgets, we refer readers to Appendix M.

### 7.2. RQ2: How Does Curvature Impact the Performance of Game-theoretic Data Values?

To empirically examine our theoretical analysis of utility curvature's impact on data valuation methods, we construct a controlled experimental framework using iterative message-passing. Given a dataset with class labels, we implement an iterative feature aggregation process parameterized by a propagation proportion $p \in [0, 1]$. At each iteration, each data point $x_i$ updates its features according to:

$$x_i^{new} = (1 - p)x_i + p \cdot \text{mean}(x_j | j \in \mathcal{N}_i)$$

where $\mathcal{N}_i$ represents the set of within-class neighbors of point $i$. This mechanism provides control over data point substitutability, which serves as a proxy for the utility function's curvature: at $p = 0$ (no message-passing), points maintain their original distinctive features (low substitutability, corresponding to low curvature), while at $p = 1$ (full message-passing), points within each class converge through complete feature averaging (high substitutability, corresponding to high curvature). Through systematic variation of $p$, we examine the impact of increasing feature similarity on both the mean performance and selection curves of different valuation methods.

Our results in Figure 3 demonstrate three key findings that are consistent with the theoretical predictions (more detailed six propagation steps with step size 0.2 analysis in Appendix K): (1) under low substitutability ($p = 0.0$), all methods `DataShap`, `BetaShap`, and `Banzhaf` achieve strong performance with mean accuracy above 0.70; (2) as substitutability increases through higher propagation propor-

*Table 1.* Performance comparison between optimal (`DynamicProgramming`) and existing methods across different datasets. We report optimal and best baseline performance, where $\Delta$ (%) shows relative performance drop from optimal.

| Dataset | 2dplanes | nomao | bbc-embeddings | MiniBooNE | digits | election | electricity | fried |
|---|---|---|---|---|---|---|---|---|
| Optimal DP | $0.741^{\pm0.082}$ | $0.842^{\pm0.131}$ | $0.698^{\pm0.165}$ | $0.753^{\pm0.068}$ | $0.177^{\pm0.213}$ | $0.549^{\pm0.227}$ | $0.681^{\pm0.059}$ | $0.736^{\pm0.072}$ |
| Best Baseline | BetaShap | DataShap | DataShap | DataShap | AME | Influence | BetaShap | Random |
| $\Delta$ (%) | -9.62% | -11.97% | -23.76% | -9.29% | -17.22% | -17.54% | -9.66% | -8.39% |

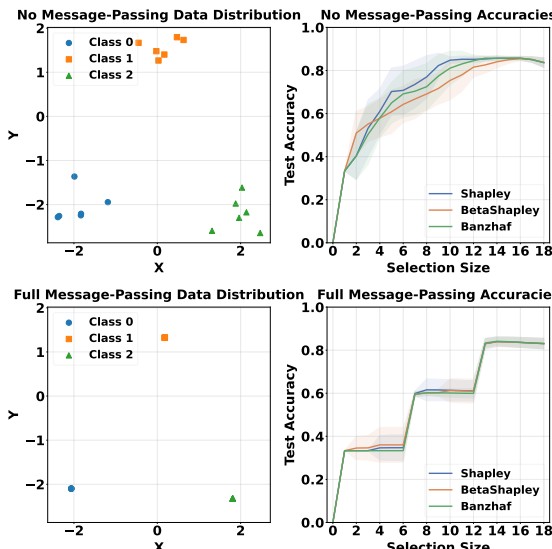

*Figure 3.* As within-class feature aggregation increases (from top to bottom with propagation proportion from 0.0 to 1.0), points become increasingly substitutable and mean accuracies decrease: Shapley (0.738→0.595), BetaShapley (0.706→0.597), and Banzhaf (0.720→0.590). Using substitutability as a proxy for curvature, this degradation pattern is consistent with the $(1-c)^2$ bound in Theorem 4.4.

tions (serving as a curvature proxy), we observe systematic performance degradation across all methods; and (3) the convergence in performance between these game-theoretic methods at high substitutability is consistent with our analysis that all these approaches face similar fundamental limitations under strong substitution effects. Note that we do not directly compute the true curvature $c$; rather, we use substitutability as a qualitative proxy that aligns with the theoretical framework.

### 7.3. RQ3: How Effective is Bipartite-based Approximation for Data Selection?

Following the experimental settings detailed in Appendix J.6, the selection curves in Figure 4 demonstrate the clear advantages of our bipartite-based approach `Bipartite`. On structured datasets like bbc-embeddings, our method exhibits remarkably steep initial performance improvements, achieving over 60% accuracy with just 20 samples, ahead of the baselines in this early-selection regime. This pattern of superior early-stage selection is consistently observed

across different datasets.

The performance advantage is particularly pronounced on complex datasets with heterogeneous feature distributions. In the digits dataset, our method demonstrates exceptional sample efficiency, reaching 80% accuracy with just 25 samples, while baseline methods require nearly twice as many samples to achieve comparable performance. Similar patterns emerge in the nomao dataset, where our method shows a clear advantage in the crucial early stages where efficient selection is most valuable, and attains the highest average accuracy over the selection curve. Even on relatively simpler datasets like 2dplanes and fried, our method shows noticeable improvements in selection efficiency. The curves reveal not only faster initial accuracy gains but also more stable progression, with fewer fluctuations compared to baseline methods. The electricity dataset results further emphasize this stability, where our method shows a strong early-stage advantage and attains the highest average accuracy. To quantify these observations, we present detailed average performance metrics for the selection curves of Figure 4 in Table 9.

**Large-scale Data Selection Effectiveness.** To further validate the effectiveness on large-scale data selection, we extended our evaluation to larger selection budgets of 500 data points. As shown in Table 2, our `Bipartite` method achieves the highest average accuracy (0.828) across all datasets. Notably, it ranks first on three datasets (bbc-embeddings, MiniBooNE, digits) and ties for first on two others (election, electricity). The performance gaps between methods narrow at larger budgets, as expected when selecting a significant portion of the dataset. Yet our method consistently maintains competitive or superior performance. Traditional methods like `DataShap` and `BetaShap` perform competitively (averaging 0.820 and 0.821, respectively) compared to learning-based approaches like `DVRL` (0.758), aligning with our theoretical analysis that game-theoretic methods maintain reliable performance when data substitutability effects diminish at larger selection sizes.

**Utility Approximation Performance.** To validate our bipartite model's effectiveness as a utility approximation, we compare its approximation quality against standard baselines including linear regression, MLP, and game-theoretic methods. We evaluate the surrogate actually used by each method on the same sampled subsets: for linear and game-theoretic baselines, this is the linear surrogate $\hat{U}(S) = \hat{b} + \sum_{i \in S} \theta_i$;

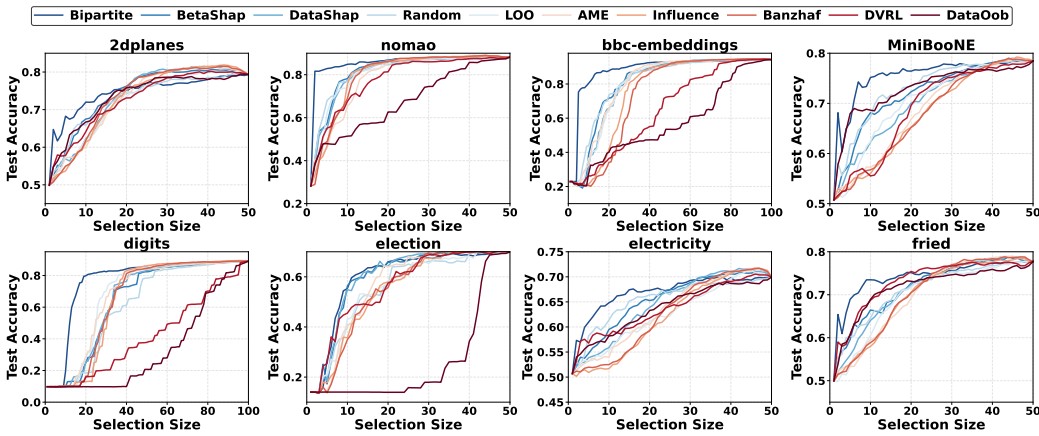

*Figure 4.* Evaluation of our proposed bipartite-based method against baselines on eight datasets.

*Table 2.* Performance comparison for large-scale data selection (budget size 500) across 20 independent runs. We report the top two performers for each dataset, with accuracy ± standard deviation. The Average column ranks methods by their per-method average over datasets rather than averaging the preceding winner cells. **Bold** indicates best performance.

| Method | 2dplanes | nomao | bbc-embeddings | MiniBooNE | digits | election | electricity | fried | Average |
|---|---|---|---|---|---|---|---|---|---|
| 1st Place Performance | DataOob $0.825^{\pm0.039}$ | DataOob $0.902^{\pm0.060}$ | Bipartite $0.946^{\pm0.076}$ | Bipartite $0.804^{\pm0.033}$ | Bipartite $0.901^{\pm0.130}$ | Bipartite* $0.715^{\pm0.078}$ | Bipartite* $0.722^{\pm0.032}$ | DataOob $0.826^{\pm0.036}$ | Bipartite **0.828** |
| 2nd Place Performance | BetaShap $0.821^{\pm0.048}$ | Random $0.899^{\pm0.050}$ | Random $0.935^{\pm0.113}$ | BetaShap $0.801^{\pm0.045}$ | Random $0.875^{\pm0.193}$ | – - | – - | Bipartite $0.820^{\pm0.035}$ | BetaShap 0.821 |

*For election: LOO, DataShap, BetaShap, and Bipartite all achieve 0.715. For electricity: DataOob and Bipartite both achieve 0.722.

for Bipartite, it is the learned coverage surrogate; and for MLP, it is the fitted nonlinear predictor. When a method is characterized by centered-feature regression (e.g., Banzhaf or Beta Shapley in Theorem 4.1), we use the equivalent uncentered parameterization from Remark D.2, and for fairness fit only the intercept $\hat{b}$ by least squares on the same sampled subsets (this does not change the induced ranking but is necessary for comparing approximation error). As shown in Table 3, our method achieves significantly lower approximation error (MSE: 0.019) compared to all baselines, with the next best method (linear regression) having more than four times higher error (MSE: 0.089). This validates our theoretical insight that the bipartite coverage structure effectively captures the underlying utility relationships while maintaining computational efficiency.

*Table 3.* Utility approximation error of each method's surrogate used for selection. Linear and game-theoretic rows use $\hat{U}(S) = \hat{b} + \sum_{i \in S} \theta_i$ with the intercept $\hat{b}$ fit by least squares; Bipartite uses its coverage surrogate and MLP uses its fitted nonlinear predictor. Lower is better.

| Method | Test | | Train | |
|---|---|---|---|---|
| | **MAE** | **MSE** | **MAE** | **MSE** |
| Bipartite | **0.102** | **0.019** | **0.104** | **0.020** |
| Linear | 0.250 | 0.089 | 0.239 | 0.083 |
| Banzhaf | 0.303 | 0.128 | 0.298 | 0.121 |
| BetaShap | 1.119 | 2.293 | 1.118 | 2.325 |
| DataShap | 1.145 | 2.401 | 1.144 | 2.431 |
| MLP | 0.301 | 0.124 | 0.307 | 0.124 |

**LLM Fine-tuning Data Selection.** Beyond classical ML benchmarks, we evaluate on large-scale LLM fine-tuning using the DATE-LM benchmark (Jiao et al., 2025). We adapt our bipartite approach to this setting as BipCov, which constructs the bipartite graph using RAG-style embeddings between pool instructions and task reference examples (details in Appendix J.7). Under the official DATE-LM pipeline (200k instruction pool, select 10k examples, LoRA fine-tune Llama-3.1-8B, evaluate on MMLU/GSM8K/BBH), BipCov achieves 63.89±0.34% average accuracy (mean±std over three seeds), improving over the strong baseline RDS+ (62.88±0.25%) and Random (62.82±0.31%). Unlike similarity-based methods that score each example independently, BipCov first prioritizes reference coverage and then falls back to reference similarity after binary coverage saturates, yielding a coverage-prioritized ordering for instruction selection.

## 8. Conclusion

We establish a theoretical framework unifying data valuation and sequential decision-making, showing that existing game-theoretic methods are myopic linear approximations with fundamental limitations under high utility curvature. To bridge the gap, we develop an effective bipartite graph-based utility approximation and data selection method.

## Acknowledgements

This research was supported by the National Science Foundation (NSF) under grant numbers NSF2406647 and NSF2406648. It was also supported by the National Artificial Intelligence Research Resource (NAIRR) Pilot and the Delta advanced computing and data resource, which is supported by the National Science Foundation under award NSF-OAC-2005572. This work was also supported by IBM through the IBM-Rensselaer Future of Computing Research Collaboration.

## Impact Statement

This paper presents work whose goal is to advance the field of machine learning. There are many potential societal consequences of our work, none of which we feel must be specifically highlighted here.

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

# A. Related Work

## A.1. Data Valuation

Data valuation seeks to quantify the contribution of individual training samples to model performance. Game-theoretic approaches have dominated this field, beginning with Data Shapley (Ghorbani & Zou, 2019), which adapts the Shapley value from cooperative game theory to quantify data contributions. This seminal work inspired various extensions, including Beta Shapley (Kwon & Zou, 2022), which introduces a family of semi-values through Beta function weighting, and Data Banzhaf (Wang & Jia, 2023), which provide robust valuation frameworks through binary-weighted marginal contributions. While these general methods are computationally intensive, specialized approaches like (Jia et al., 2019a) achieve near-linear time complexity by exploiting algorithmic structures such as K-Nearest Neighbors, compared to the exponential complexity of model-agnostic approaches. Recent developments address specific challenges in data valuation: (Schoch et al., 2022) proposed CS-Shapley for better handling class-wise contributions in classification tasks, while (Chi et al., 2024) introduced PC-Winter value to tackle the unique challenges of graph-structured data valuation. Alternative theoretical frameworks, such as the Core (Yan & Procaccia, 2021), have also emerged to address coalition stability in data valuation.

Recent work has focused on improving computational efficiency and valuation accuracy. (Wang et al., 2021) proposed learning data utility functions to avoid repeated model retraining, while (Garrido-Lucero et al., 2023) developed DU-Shapley as an efficient proxy through discrete uniform distributions. (Tarun et al., 2024) introduced EcoVal, which accelerates valuation by clustering similar data points and propagating values within clusters. P-Shapley (Xia et al., 2024a) leverages predicted probabilities instead of accuracy for finer-grained utility differentiation. For applications requiring only top-ranked data points, (Lin et al., 2025) proposed GPGapE, which uses Gaussian processes to model non-linear mappings from data features to values, significantly reducing the computational burden for top-$m$ data identification. In the graph domain, (Chi et al., 2025) introduced Shapley-Guided Utility Learning for effective graph inference data valuation, combining transferable data-specific and model-specific features to approximate test accuracy without relying on ground truth labels.

Several approaches have tackled computational challenges through gradient-based methods. (Wang et al., 2024b) introduced In-Run Data Shapley, which calculates Shapley values during model training without requiring retraining, enabling data attribution even for large foundation models. Building on this, (Wang et al., 2024d) proposed data value embedding to capture temporal dependence in data influence, revealing that training data impact varies significantly across different training phases. (Covert et al., 2024) presented stochastic amortization as a unified approach to accelerate feature and data attribution, using noisy labels to train amortized models that significantly accelerate several data valuation methods.

Parallel efforts have explored training-free and task-agnostic directions. (Kessler et al., 2024) developed SAVA, a scalable variant of the LAVA framework (Just et al., 2023) that applies optimal transport between training and validation sets in batches rather than requiring the entire dataset. For generative models, (Yang et al., 2023) introduced GMValuator, a similarity-based approach that evaluates training data contributions to generated samples without model retraining. (Tong et al., 2025) proposed a fine-grained analysis called Dataset Usage Cardinality Inference to estimate the exact proportion of data used in training, addressing security concerns for data owners assessing unauthorized usage risks. In the context of large language models, (Gu et al., 2024) formulated data selection as a generalized Optimal Control problem and introduced PMP-based Data Selection for accelerating the learning of language models through optimally selected pre-training data.

Additional notable approaches include DAVINZ (Wu et al., 2022), which enables data valuation at network initialization by theoretically deriving domain-aware generalization bounds, while (Amiri et al., 2023) proposed a task-agnostic framework based on statistical properties without validation requirements. Alternative paradigms include reinforcement learning-based approaches like DVRL (Yoon et al., 2020) and complexity-gap scoring (Nohyun et al., 2023), which offer diverse perspectives beyond traditional game-theoretic methods.

## A.2. Approximate Dynamic Programming

Sequential decision-making under uncertainty has been extensively studied through the lens of Markov Decision Processes (Bellman, 1966). However, exact solutions become computationally intractable for large state spaces, motivating the development of Approximate Dynamic Programming (ADP) methods (Powell, 2007; Bertsekas, 2024). Key ADP techniques include value function approximation, which represents the optimal value function through parameterized function classes, and policy gradient methods, which directly optimize parameterized policies (Sutton & Barto, 2018). Recent advances combine these approaches with deep learning, leading to successful applications in game playing (Silver et al., 2016) and robotics (Levine et al., 2016).

The connection between combinatorial optimization and sequential decision-making has been explored through submodular function maximization, where greedy algorithms provide constant-factor approximations (Nemhauser et al., 1978). Our work bridges these areas by showing that data valuation methods implicitly solve a sequential selection problem, with their approximation quality governed by utility function curvature.

### A.3. Semi-values

Semi-values generalize the Shapley value by relaxing the efficiency axiom while maintaining other desirable properties (Weber, 1988). The Banzhaf value (Banzhaf III, 1965) was originally developed for analyzing voting power, while Beta Shapley (Kwon & Zou, 2022) provides a flexible family of semi-values parameterized by Beta function weights. Recent theoretical work has characterized semi-values through weighted least squares regression (Charnes et al., 1988), a perspective we leverage to establish the ADP interpretation of data valuation methods.

### A.4. Submodular Optimization

Submodular functions provide a powerful mathematical framework for modeling diminishing returns across diverse domains (Fujishige, 2005). The seminal work by Nemhauser et al. (Nemhauser et al., 1978; Nemhauser & Wolsey, 1978) established that a greedy algorithm achieves a $(1 - 1/e)$-approximation for monotone submodular maximization under cardinality constraints—a bound that is provably tight. Recent advances have addressed more challenging settings: (Buchbinder et al., 2014) improved approximation algorithms for non-monotone submodular maximization with cardinality constraints, while their subsequent work (Buchbinder et al., 2015) provided a tight $1/2$-approximation for unconstrained submodular maximization. Researchers have also explored various extensions, including weak submodularity (Santiago & Yoshida, 2020), submodular cover and knapsack constraints (Iyer & Bilmes, 2013), and probabilistic models defined through submodular functions (Djolonga & Krause, 2014). Our work draws on these theoretical foundations, particularly the analysis of approximation bounds, to develop efficient algorithms for data selection that maintain theoretical guarantees while improving computational tractability.

Another closely related domain of sequential submodular optimization (Asadpour et al., 2023; Tang & Yuan, 2024; Zhang et al., 2022) focuses on optimizing over a sequence of different submodular functions, particularly targeting applications like product ranking in recommendation systems. Unlike these works which optimize across multiple different submodular functions (where each function evaluates prefixes of different lengths), our Sequential Data Selection Problem maximizes the expected utility across different selection sizes using a single consistent utility function.

### A.5. Optimization from Samples

Optimization from Samples (OPS) addresses how to optimize functions when only sample evaluations are available, not direct oracle access (Balkanski et al., 2017a). This paradigm reveals a fundamental gap between learnability and optimizability: (Balkanski et al., 2016) showed that for submodular functions with bounded curvature, approximation algorithms exist using polynomial samples, yet (Balkanski et al., 2017a) established that for coverage functions, no constant-factor approximation is possible with polynomial samples from any distribution. To overcome these limitations, (Chen et al., 2020) introduced Optimization from Structured Samples (OPSS), where samples encode structural information about the function. With assumptions on subset coverage patterns and marginal contributions, they achieved constant approximations for coverage maximization. For submodular functions, curvature critically affects approximation guarantees. (Balkanski et al., 2016) established a $(1 - c)/(1 + c - c^2)$-approximation for submodular functions with curvature $c$, which aligns with our analysis in Section 4.1 where we prove data valuation methods achieve a $(1 - c)^2$ approximation for sequential selection. Our bipartite graph approximation method draws inspiration from these structural approaches. Also, (Balkanski et al., 2017b) extends these sample-based approaches to cooperative games where cost functions must be learned from samples rather than accessed directly.

### A.6. Other Linear Surrogate Models for Data Selection

Beyond traditional data valuation methods based on cooperative game theory (Ghorbani & Zou, 2019; Wang & Jia, 2023; Kwon & Zou, 2022), recent approaches have explored linear surrogate functions for model-aware data selection. (Ilyas et al., 2022) introduced datamodels, a framework that maps training subsets to model predictions through linear functions, while (Engstrom et al., 2024) extended this approach with DsDm to directly optimize training data selection for target tasks. While these methods also model relationships between training data and model performance, our work uniquely frames data

selection as a sequential decision process with theoretical guarantees under varying utility structures.

### A.7. Efficient Data Valuation Methods

Recent work has explored various approaches to address the computational challenges in data valuation. These methods can be broadly categorized into two complementary directions. Gradient-based acceleration methods such as G-Shapley (Ghorbani & Zou, 2019), All-S Influence (Jia et al., 2019b), and FreeShap (Wang et al., 2024a) focus on reducing the computational cost of each individual model training through gradient approximations or parameter sharing techniques. These approaches maintain the full valuation framework while making each utility evaluation more efficient. In contrast, our Bipartite method belongs to a different category that aims to reduce the total number of model evaluations needed by learning more effective approximations of the utility landscape. While gradient-based methods accelerate each utility computation instance, our approach requires fewer total evaluations to achieve superior selection performance. These two categories of methods are conceptually complementary and could potentially be combined; using gradient-based acceleration for the subset evaluations required by our bipartite graph construction process would further enhance computational efficiency while maintaining the theoretical guarantees of our framework.

### A.8. Parallel Work on Unifying Data Selection Methods

Recent work by (Wang et al., 2024c) presents a comprehensive tutorial on data selection methods for foundation models, focusing on both heuristic and principled approaches to data curation. While their work provides valuable insights into the practical aspects of data selection in foundation model training pipelines, our work differs in several key aspects: First, we focus specifically on the optimization perspective of data selection with data values, providing theoretical guarantees of optimality under our proposed sequential decision making and approximate dynamic programming framework. Their work takes a broader view, covering various selection strategies without explicit optimization guarantees. Second, our work presents a novel unifying framework via approximate dynamic programming (ADP) (Bertsekas, 2024). This framework reveals that many existing data valuation methods can be interpreted as specific instances of myopic cost function approximation heuristic (Powell, 2016; Rempel & Cai, 2021), which is a classical strategy in the ADP literature. This theoretical connection not only provides new insights into why existing methods work but also suggests principled ways to improve them. Third, while both works aim to bridge the gap between heuristic and principled approaches, we focus more on the theoretical foundations of data valuation methods for selection tasks. We provide rigorous analysis of the limitations of data attribution methods and characterize the conditions under which they can achieve optimal performance. Our work thus complements theirs by providing deeper theoretical understanding of data valuation based selection methods, while their tutorial offers broader practical guidance across different selection paradigms. Together, these parallel efforts contribute to advancing both the theoretical foundations and practical applications of data selection methods. In parallel, GraphFilter (Wu et al., 2025) also formulates data selection over a bipartite graph, but with a sentence–n-gram construction and a set-cover greedy procedure aimed at LLM SFT diversity, in contrast to our training–validation utility-coverage surrogate analyzed within the sequential decision-making framework.

## B. Complete Formulation of Theorem 4.1

This section provides the complete mathematical formulations for the regression characterizations of game-theoretic data values stated in Theorem 4.1.

**Part (A) [Shapley: Constrained WLS (KernelSHAP form)]:** Assume $n = |\mathcal{D}| \geq 2$. The Shapley value $v_{\text{Shap}}(i)$ is the unique solution to the constrained weighted least squares problem:

$$\min_{\theta \in \mathbb{R}^n} \sum_{\substack{S \subseteq \mathcal{D} \\ 1 \leq |S| \leq n-1}} \omega_{\text{Shap}}(|S|)\Big(U(S) - \sum_{i \in S} \theta_i\Big)^2 \quad \text{s.t.} \quad \sum_{i \in \mathcal{D}} \theta_i = U(\mathcal{D}), \tag{3}$$

where the size weight is $\omega_{\text{Shap}}(k) = \binom{n-2}{k-1}^{-1}$ for $1 \leq k \leq n-1$. The sum excludes $|S| = 0$ and $|S| = n$ since these boundary cases are handled by the normalization $U(\emptyset) = 0$ and the efficiency constraint.

**Part (B1) [Banzhaf: Unconstrained OLS with Centered Features]:** Sample $S \sim \text{Unif}(2^{\mathcal{D}})$ (each element included i.i.d.

with $p = \frac{1}{2}$). Define centered features $z_i(S) = \mathbb{I}_{i \in S} - \frac{1}{2}$. Then the Banzhaf value satisfies:

$$v_{\text{Banz}}(i) = \theta_i^*, \quad (\theta^*, b^*) = \arg\min_{b, \theta} \ \mathbb{E}_S\left[\left(U(S) - b - \sum_{i \in \mathcal{D}} \theta_i z_i(S)\right)^2\right]. \tag{4}$$

**Part (B2) [Beta Shapley: Weighted OLS with Centered Features]:** Assume $a > 1$ and $b > 1$, and assume $U$ is bounded (see Remark D.1 for well-definedness conditions). Draw $p \sim \text{Beta}(a, b)$; then $S \mid p \sim \text{Bern}(p)^{\mathcal{D}}$. Define $z_i(S, p) = \mathbb{I}_{i \in S} - p$ and weight $w(p) = 1/[p(1-p)]$. The Beta Shapley value satisfies:

$$v_{\text{Beta}}(i) = \theta_i^*, \quad (\theta^*, b^*) = \arg\min_{b, \theta} \ \mathbb{E}_{p, S}\left[w(p)\left(U(S) - b - \sum_{i \in \mathcal{D}} \theta_i z_i(S, p)\right)^2\right]. \tag{5}$$

*Table 4.* Population regression characterizations via subset sampling and centered features

| Value | Subset sampling | Regression form |
|---|---|---|
| Shapley | Size-weights $\omega_{\text{Shap}}(\lvert S\rvert)$ | $\min_\theta \sum_S \omega_{\text{Shap}}(\lvert S\rvert)\left(U(S) - \sum_{i \in S}\theta_i\right)^2$
s.t. $\sum_i \theta_i = U(\mathcal{D})$ |
| Banzhaf | $S \sim \text{Unif}(2^{\mathcal{D}})$
(i.i.d. Bern$(1/2)$) | $\min_{b,\theta} \ \mathbb{E}\left(U(S) - b - \sum_i \theta_i(\mathbb{I}_{i \in S} - \frac{1}{2})\right)^2$

(no efficiency constraint) |
| BetaShap$(a, b)$ | $p \sim \text{Beta}(a, b)$
$S\lvert p \sim \text{Bern}(p)^{\mathcal{D}}$ | $\min_{b,\theta} \ \mathbb{E}\left[\frac{1}{p(1-p)}\left(U(S) - b - \sum_i \theta_i(\mathbb{I}_{i \in S} - p)\right)^2\right]$

(no efficiency constraint; $a, b > 1$) |

*Remark* B.1 (Why Different Regression Forms?). The efficiency constraint $\sum_i \theta_i = U(\mathcal{D})$ in Part (A) reflects Shapley's efficiency axiom. Banzhaf and Beta Shapley violate efficiency for general $U$, so imposing this constraint would *not* recover them. Instead, Parts (B1) and (B2) use unconstrained regression with centered features; the intercept $b$ absorbs the "slack" from the missing efficiency property.

## C. Proof of Theorem 4.1

We provide proofs for all three parts of Theorem 4.1.

### C.1. Part (A): Shapley as Constrained WLS

*Proof.* The KernelSHAP regression characterization of Shapley values is well-established (Lundberg & Lee, 2017; Charnes et al., 1988). For completeness, we sketch the key steps.

The Shapley value can be written as:

$$v_{\text{Shap}}(i) = \sum_{S \subseteq \mathcal{D} \setminus \{i\}} \frac{\lvert S\rvert!(n - \lvert S\rvert - 1)!}{n!} \Delta_i U(S)$$

Consider the constrained WLS problem in Equation (3). The Lagrangian is:

$$\mathcal{L}(\theta, \lambda) = \sum_S \omega(\lvert S\rvert)\left(U(S) - \sum_{i \in S}\theta_i\right)^2 + \lambda\left(\sum_i \theta_i - U(\mathcal{D})\right)$$

Taking derivatives and solving the first-order conditions with the KernelSHAP weights $\omega(k) = \binom{n-2}{k-1}^{-1}$ yields $\theta_i = v_{\text{Shap}}(i)$.

**Note.** The KernelSHAP size-weight admits equivalent forms up to a global scaling constant. In particular,

$$\frac{1}{\binom{n-2}{k-1}} = \frac{n(n-1)}{k(n-k)} \cdot \frac{1}{\binom{n}{k}},$$

so $\omega_{\text{Shap}}(k) = \binom{n-2}{k-1}^{-1}$ is equivalent (up to scaling) to the commonly used kernel $\propto 1/(\binom{n}{k}k(n-k))$ for recovering Shapley coefficients. $\qquad\square$

## C.2. Part (B1): Banzhaf as Unconstrained OLS

*Proof.* Sample $S \sim \text{Unif}(2^{\mathcal{D}})$ by including each element i.i.d. with probability $p = 1/2$.

**Step 1: Centering and orthogonality.** Define $z_i(S) = \mathbb{I}_{i \in S} - 1/2$. Then:

- $\mathbb{E}[z_i(S)] = 0$ (centered)
- $\mathbb{E}[z_i(S)z_j(S)] = 0$ for $i \neq j$ (uncorrelated)
- $\mathbb{E}[z_i(S)^2] = 1/4$ (variance)

**Step 2: OLS normal equations.** The OLS solution minimizes $\mathbb{E}[(U(S) - b - \sum_i \theta_i z_i(S))^2]$. By the first-order conditions:

$$\theta_i = \frac{\mathbb{E}[U(S)z_i(S)]}{\mathbb{E}[z_i(S)^2]} = 4\mathbb{E}[U(S)(\mathbb{I}_{i \in S} - 1/2)]$$

**Step 3: Identification with Banzhaf.**

$$\mathbb{E}[U(S)(\mathbb{I}_{i \in S} - 1/2)] = \mathbb{E}[U(S)\mathbb{I}_{i \in S}] - \frac{1}{2}\mathbb{E}[U(S)]$$

$$= \frac{1}{2^n} \sum_{S \ni i} U(S) - \frac{1}{2^{n+1}} \sum_S U(S)$$

After simplification using the Banzhaf formula $v_{\text{Banz}}(i) = \frac{1}{2^{n-1}} \sum_{S \subseteq \mathcal{D} \setminus \{i\}} \Delta_i U(S)$, we obtain $\theta_i = v_{\text{Banz}}(i)$. $\qquad\square$

## C.3. Part (B2): Beta Shapley as Weighted OLS

*Proof.* Draw $p \sim \text{Beta}(a, b)$ with $a > 1, b > 1$, then $S|p \sim \text{Bern}(p)^{\mathcal{D}}$.

**Step 1: Conditional expectation.** For a fixed $p$ and $T = S \setminus \{i\}$:

$$\mathbb{E}[U(S)(\mathbb{I}_{i \in S} - p)|p, T] = p(1-p)\big[U(T \cup \{i\}) - U(T)\big] = p(1-p)\Delta_i U(T)$$

**Step 2: Effect of weight $w(p) = \frac{1}{p(1-p)}$.** The weighted normal equations give:

$$\theta_i = \frac{\mathbb{E}[w(p)U(S)z_i(S, p)]}{\mathbb{E}[w(p)z_i(S, p)^2]}$$

The denominator: $\mathbb{E}[w(p) \cdot p(1-p)] = \mathbb{E}[1] = 1$.

The numerator:

$$\mathbb{E}\left[\frac{1}{p(1-p)} \cdot p(1-p)\Delta_i U(T)\right] = \mathbb{E}_p \mathbb{E}_{T|p}[\Delta_i U(T)]$$

Therefore $\theta_i = \mathbb{E}_{p \sim \text{Beta}(a,b)} \mathbb{E}_{T \sim \text{Bern}(p)^{\mathcal{D} \setminus \{i\}}}[\Delta_i U(T)]$, which is the Beta Shapley value. $\qquad\square$

# D. Technical Remarks on Regression Characterizations

This section collects technical remarks that support the main results in Section 4.

*Remark* D.1 (Well-definedness of the weighted objective). The assumption $a > 1, b > 1$ in Part (B2) of Theorem 4.1 ensures the population weighted least-squares objective with $w(p) = 1/[p(1-p)]$ is finite, hence the normal equations are well-defined. Additionally, we assume $U$ is bounded (e.g., $U(S) \in [0, 1]$ when $U$ represents accuracy), which ensures the objective $\mathbb{E}[w(p)(U(S) - \cdots)^2]$ has finite second moments; this is satisfied in typical data valuation applications where $U$ is a bounded performance metric. If one wishes to allow $a, b \leq 1$, one may instead use a truncated weight $w_\varepsilon(p) = 1/[(p \vee \varepsilon)(1 - p \vee \varepsilon)]$ and take $\varepsilon \to 0$. The characterization is with respect to the joint sampling process $p \sim \text{Beta}(a, b)$ and $S|p \sim \text{Bern}(p)^{\mathcal{D}}$, i.e., $\theta^*$ is the population minimizer under this generative model. Note that the special case $a = b = 1$ corresponds to $p \sim \text{Uniform}(0, 1)$, for which the *Beta Shapley value itself* coincides with the standard Shapley value (hence satisfies efficiency $\sum_i v(i) = U(\mathcal{D}) - U(\emptyset)$). However, our regression characterization uses the weight $w(p) = 1/[p(1-p)]$, for which $\mathbb{E}[w(p)] = \infty$ under $p \sim \text{Uniform}(0, 1)$. Therefore the regression form is well-defined only for $a > 1, b > 1$ as stated. To cover $a = b = 1$ within a regression view, one may use a truncated weight and interpret the result in the limit $\varepsilon \to 0$, or alternatively view Shapley as the constrained WLS form in Part (A).

*Remark* D.2 (Reparameterization for Banzhaf and selection invariance for Beta Shapley). **Banzhaf.** Because the centering constant $\mu = \frac{1}{2}$ is deterministic, the centered-feature linear model is exactly equivalent to a raw-indicator linear surrogate after absorbing the constant term into the intercept:

$$b + \sum_{i \in \mathcal{D}} \theta_i (\mathbb{I}_{i \in S} - \tfrac{1}{2}) = b' + \sum_{i \in \mathcal{D}} \theta_i \mathbb{I}_{i \in S}, \qquad b' \triangleq b - \tfrac{1}{2} \sum_{i \in \mathcal{D}} \theta_i.$$

**Beta Shapley.** The fitted regression model retains the latent sampling variable $p$ through the additive term $-p \sum_i \theta_i$, so there is no fixed-intercept set function of $S$ alone that is literally equal to the population fit. However, this term is independent of the subset action: for any fixed $p$ and any $a \notin S$,

$$\hat{U}(S \cup \{a\}, p) - \hat{U}(S, p) = \theta_a.$$

Hence the myopic policy at any state depends only on the coefficients $\theta_i$, even though the regression objective is not literally identical to a surrogate function of $S$ alone. This action-invariance of the marginal gain is what Theorem 4.2 relies on.

*Remark* D.3 (Scope of the Regression Characterization). The regression characterizations in Theorem 4.1 apply specifically to *semi-values* (Shapley, Banzhaf, Beta Shapley) where the weights $\alpha_i(S) = \alpha_{|S|}$ depend only on subset size, not on the element $i$ being valued. For general game-theoretic values with $i$-dependent weights $\alpha_i(S)$, a natural population regression characterization may not exist. In particular, LOO values, which use $\alpha_i(S) = 1$ if $S = \mathcal{D} \setminus \{i\}$ and 0 otherwise, do not arise from any standard regression objective. Thus, while our ADP framework "unifies" existing semi-value methods through the regression plus myopic-policy lens, this interpretation does not extend to all possible marginal-contribution weightings.

**Clarification on LOO values.** LOO values $v_{\text{LOO}}(i) = U(\mathcal{D}) - U(\mathcal{D} \setminus \{i\})$ induce a ranking but do *not* arise as regression coefficients of any natural linear surrogate. Specifically, there is no population regression problem whose solution $\hat{\theta}$ satisfies $\hat{\theta}_i = v_{\text{LOO}}(i)$ in general. One could artificially *construct* a linear function $\hat{U}(S) = \hat{b} + \sum_i v_{\text{LOO}}(i) \cdot \mathbb{I}_{i \in S}$, but this is not derived from fitting $U$ via any standard regression objective. Therefore, while LOO values can be used for ranking (and fall within the curvature guarantee of Theorem 4.4), they do not fit the "regression $\rightarrow$ myopic ADP" interpretation that unifies Shapley/Banzhaf/Beta Shapley.

# E. Proof of Game-theoretic Data Values as ADP Solutions

We provide the complete proof that data value-induced rankings correspond to myopic ADP trajectories.

**Theorem E.1** (Restatement of Theorem 4.2). *Let $v : \mathcal{D} \rightarrow \mathbb{R}$ be a data value that admits a linear surrogate $\hat{U}(S) = \hat{b} + \sum_{i \in S} \theta_i$ with $\theta_i = v(i)$ (e.g., the Shapley, Banzhaf, or Beta-Shapley values of Theorem 4.1). Then the ranking sequence $\pi_v$ induced by $v$ is a trajectory of the ADP framework with:*

1. *Linear surrogate reward: $\hat{U}(S) = \hat{b} + \sum_{i \in S} \theta_i$ where $\theta_i = v(i)$*
2. *Myopic policy: $\pi_{myopic}(s) = \arg\max_{a \in \mathcal{D} \setminus s} \theta_a$*

*Proof.* The proof proceeds in three steps.

**Step 1: Linear surrogate marginal gains are constant.** For the linear surrogate $\hat{U}(S) = \hat{b} + \sum_{i \in S} \theta_i$:

$$\hat{U}(S \cup \{a\}) - \hat{U}(S) = \theta_a$$

This marginal gain is independent of both the current state $S$ and the intercept $\hat{b}$.

**Step 2: Myopic policy induces value-based ranking.** The myopic policy at any state $s$ selects:

$$\pi_{myopic}(s) = \arg\max_{a \in \mathcal{D} \setminus s} [\hat{U}(s \cup \{a\}) - \hat{U}(s)] = \arg\max_{a \in \mathcal{D} \setminus s} \theta_a$$

Since $\theta_a$ is state-independent, starting from $s_0 = \emptyset$:

- Step 1: Select $a_1 = \arg\max_{a \in \mathcal{D}} \theta_a$
- Step 2: Select $a_2 = \arg\max_{a \in \mathcal{D} \setminus \{a_1\}} \theta_a$
- Step $t$: Select $a_t = \arg\max_{a \in \mathcal{D} \setminus \{a_1, \ldots, a_{t-1}\}} \theta_a$

This is precisely the sequence of elements sorted in descending order of $\theta$.

**Step 3: Identification with data values.** By Theorem 4.1, for game-theoretic data values:

$$\theta_i = v(i)$$

Therefore, the myopic trajectory $(a_1, a_2, \ldots, a_n)$ corresponds to the permutation $\pi_v$ that sorts elements by their data values, completing the proof. □

## F. Proof of Linear Utility Optimality

**Lemma F.1** (Consistent Marginal Contributions). *For a linear utility function $U(S) = \sum_{i \in S} w_i$, the marginal contribution of any element $i$ remains constant regardless of the subset $S$:*

$$U(S \cup \{i\}) - U(S) = w_i, \quad \forall S \subseteq \mathcal{D} \setminus \{i\}$$

**Lemma F.2** (Semi-value Equivalence). *Under a linear utility function, any semi-value based method assigns values equal to the weights:*

$$v(i) = w_i$$

*Proof.* For any semi-value based method:

$$v(i) = \sum_{S \subseteq \mathcal{D} \setminus \{i\}} \alpha_{|S|}[U(S \cup \{i\}) - U(S)]$$

$$= \sum_{S \subseteq \mathcal{D} \setminus \{i\}} \alpha_{|S|} w_i$$

$$= w_i \sum_{S \subseteq \mathcal{D} \setminus \{i\}} \alpha_{|S|} = w_i$$

where $\sum_{S \subseteq \mathcal{D} \setminus \{i\}} \alpha_{|S|} = \sum_{k=0}^{n-1} \binom{n-1}{k} \alpha_k = 1$ by the semi-value normalization (Definition 2.3). □

**Theorem F.3** (Sequential Selection Optimality). *For any linear utility function $U$, selecting elements in descending order of their semi-values achieves optimal cumulative utility:*

$$\sum_{k=1}^{n} U(S_k^v) = \max_{\pi} \sum_{k=1}^{n} U(S_k^\pi)$$

*Proof.* The proof follows in three steps:

1) By Lemma F.2, the ordering based on semi-values $v(i)$ is equivalent to ordering based on weights $w_i$.

2) For any size $k$, the subset $S_k^v$ selected by this ordering contains the $k$ elements with largest weights.

3) Due to linearity, for any size $k$:

$$U(S_k^v) = \sum_{i \in S_k^v} w_i \geq \sum_{i \in S_k^\pi} w_i = U(S_k^\pi)$$

for any alternative subset $S_k^\pi$ of size $k$. Therefore:

$$\sum_{k=1}^{n} U(S_k^v) \geq \sum_{k=1}^{n} U(S_k^\pi)$$

for any alternative sequence $\pi$, establishing optimality. □

This proof shows that all semi-value methods achieve identical selection performance under linear utilities because they preserve the ordering of the underlying weights $w_i$. This explains why different methods like Data Shapley, Beta Shapley, and Data Banzhaf all achieve optimal sequential selection despite their different weighting schemes.

# G. Analysis of Data Valuation Methods Under Submodular Functions

In this appendix, we provide a complete proof that ranking-based element selection using any standard data valuation method achieves a $(1-c)^2$ approximation ratio for monotone submodular functions with curvature $c$. This result unifies previous analyses and provides general curvature-dependent bounds for Shapley value, Banzhaf value, and Leave-one-out methods, providing theoretical foundations for the results presented in Section 4.1 of the main paper.

**Standing assumptions for this appendix.**   Throughout this section, we assume:

- $U : 2^{\mathcal{D}} \to \mathbb{R}_{\geq 0}$ is **non-negative**;
- $U$ is **normalized**: $U(\emptyset) = 0$;
- $U$ is **monotone**: $U(A) \leq U(B)$ for all $A \subseteq B$;
- $U$ is **submodular**: $\Delta_i U(A) \geq \Delta_i U(B)$ for all $A \subseteq B$ and $i \notin B$;
- $U(\{i\}) > 0$ for all $i \in \mathcal{D}$ (so that curvature is well-defined).

The data value $v(i)$ is assumed to be a **non-negative normalized average of marginal contributions**:

$$v(i) = \sum_{S \subseteq \mathcal{D} \setminus \{i\}} \alpha_i(S) \Delta_i U(S), \quad \text{with } \alpha_i(S) \geq 0 \text{ and } \sum_S \alpha_i(S) = 1.$$

This includes Shapley, Banzhaf, Beta Shapley, and LOO as special cases, but excludes methods that can produce negative scores or are not representable as such normalized averages.

## G.1. Preliminaries and Submodular Functions

**Definition G.1** (Optimal Subsets)**.**  Given a utility function $U : 2^{\mathcal{D}} \to \mathbb{R}$, we define two notions of optimality:

- **Unconstrained optimal**: $OPT_k^* = \arg\max_{|S|=k} U(S)$, the best size-$k$ subset without sequential constraints
- **Sequential optimal**: $OPT_k^{\pi}$, the size-$k$ prefix of the optimal sequence solving Problem 3.1

Note that $U(OPT_k^*) \geq U(OPT_k^{\pi})$ always holds, since unconstrained optimization is at least as good as constrained.

Consider a dataset $\mathcal{D}$ and a monotone submodular utility function $U : 2^{\mathcal{D}} \to \mathbb{R}_{\geq 0}$ with $U(\emptyset) = 0$. For any sets $A \subseteq B \subseteq \mathcal{D}$ and element $i \in \mathcal{D} \setminus B$, monotone submodularity implies:

$$U(A) \leq U(B) \tag{monotonicity}$$
$$U(A \cup \{i\}) - U(A) \geq U(B \cup \{i\}) - U(B) \tag{submodularity}$$

A fundamental property of submodular functions is their curvature, which measures how much marginal contributions decrease as sets grow. For a normalized ($U(\emptyset) = 0$) monotone submodular function $U$, its curvature $c \in [0,1]$ is defined as:

$$c = 1 - \min_{i \in \mathcal{D}} \frac{U(\mathcal{D}) - U(\mathcal{D} \setminus \{i\})}{U(\{i\})}$$

This definition implies that for any element $i$ and set $S \subseteq \mathcal{D} \setminus \{i\}$:

$$U(S \cup \{i\}) - U(S) \geq (1-c)U(\{i\})$$

## G.2. Data Valuation Methods and Curvature

We consider data valuation methods that assign a value $v(i)$ to each element $i \in \mathcal{D}$ as a weighted average of marginal contributions:

$$v(i) = \sum_{S \subseteq \mathcal{D} \setminus \{i\}} \alpha(S)[U(S \cup \{i\}) - U(S)]$$

where marginal-contribution weights $\alpha(S) \geq 0$ satisfy $\sum_S \alpha(S) = 1$. A particularly important class of such valuations are semi-values, which satisfy additional symmetry properties through size-based weights:

$$\alpha(S) = \alpha_{|S|} \text{ for some } \alpha_k \geq 0 \text{ with } \sum_{k=0}^{|\mathcal{D}|-1} \binom{|\mathcal{D}|-1}{k} \alpha_k = 1$$

where $\alpha_k$ represents the weight assigned to all subsets of size $k$.

This framework encompasses several important cases from the main paper:

1. **Data Shapley**: $\alpha(S) \propto \frac{|S|!(|\mathcal{D}|-|S|-1)!}{|\mathcal{D}|!}$

2. **Data Banzhaf**: $\alpha(S) = \frac{1}{2^{|\mathcal{D}|-1}}$

3. **Leave-one-out**: $\alpha(S) = 1$ if $S = \mathcal{D} \setminus \{i\}$, 0 otherwise

We now establish a fundamental relationship between data values and singleton utility values under curvature constraints.

**Lemma G.2** (Value-Curvature Relationship). *For any element $i \in \mathcal{D}$, the data value $v(i)$ and singleton value $U(\{i\})$ satisfy the double-sided inequality:*

$$(1-c)U(\{i\}) \leq v(i) \leq U(\{i\})$$

*Proof.* **Upper bound:** By submodularity, for any $S \subseteq \mathcal{D} \setminus \{i\}$:

$$U(S \cup \{i\}) - U(S) \leq U(\{i\}) - U(\emptyset) = U(\{i\})$$

where we use $\Delta_i U(\emptyset) = U(\{i\})$ by the singleton-marginal equivalence (Remark 2.1). Since $v(i) = \sum_S \alpha(S)[U(S \cup \{i\}) - U(S)]$ with $\sum_S \alpha(S) = 1$ and all weights non-negative:

$$v(i) \leq \sum_S \alpha(S)U(\{i\}) = U(\{i\})$$

**Lower bound:** The proof follows from the curvature property and the structure of data valuations. By the curvature definition, we have $\Delta_i U(\mathcal{D} \setminus \{i\}) \geq (1-c)U(\{i\})$. By submodularity, for any $S \subseteq \mathcal{D} \setminus \{i\}$:

$$\Delta_i U(S) \geq \Delta_i U(\mathcal{D} \setminus \{i\}) \geq (1-c)U(\{i\})$$

The data value $v(i)$ is defined as a weighted average of such marginal contributions:

$$v(i) = \sum_{S \subseteq \mathcal{D} \setminus \{i\}} \alpha(S)[U(S \cup \{i\}) - U(S)]$$

Since each term in the sum is lower bounded by $(1-c)U(\{i\})$ and the weights $\alpha(S)$ are non-negative and sum to 1, we have:

$$\begin{aligned} v(i) &= \sum_{S \subseteq \mathcal{D} \setminus \{i\}} \alpha(S)[U(S \cup \{i\}) - U(S)] \\ &\geq \sum_{S \subseteq \mathcal{D} \setminus \{i\}} \alpha(S)[(1-c)U(\{i\})] \\ &= (1-c)U(\{i\}) \sum_{S \subseteq \mathcal{D} \setminus \{i\}} \alpha(S) \\ &= (1-c)U(\{i\}) \end{aligned}$$

This bound is tight for certain submodular functions and plays a crucial role in establishing our approximation guarantees. □

### G.3. Approximation Analysis

Inspired by the curvature-based analysis framework introduced by (Balkanski et al., 2016), we begin by establishing a fundamental relationship between the value of a set constructed through sequential selection and the individual values of its elements. This relationship forms the cornerstone of our approximation guarantee.

**Lemma G.3** (Sequential Selection Bound). *For any sequence of elements $e_1, \ldots, e_k$ and their corresponding partial sets $S_i = \{e_1, \ldots, e_i\}$ with $S_0 = \emptyset$, we have:*

$$U(S_k) = \sum_{i=1}^{k} \Delta_{e_i} U(S_{i-1}) \geq (1-c) \sum_{i=1}^{k} U(\{e_i\}) \geq (1-c) \sum_{i=1}^{k} v(e_i)$$

*Proof.* The first equality expresses the value of set $S_k$ as the sum of marginal contributions when elements are added sequentially (telescoping sum). For the first inequality, we apply the curvature property to each term: $\Delta_{e_i} U(S_{i-1}) \geq (1-c)U(\{e_i\})$. The second inequality follows from the upper bound in Lemma G.2, which shows that $v(e_i) \leq U(\{e_i\})$ for each element $e_i$. □

This theorem formalizes the approximation guarantee mentioned in Section 4.1 of the main paper:

**Theorem G.4** (Submodular Approximation). *Let $U$ be a monotone submodular function with curvature $c$. Let $S_k^v$ be the set of $k$ elements selected by choosing elements in decreasing order of their data values $v(\cdot)$. Then:*

$$U(S_k^v) \geq (1-c)^2 U(OPT_k^*)$$

*where $OPT_k^*$ is an optimal solution to $\max_{|S| \leq k} U(S)$ without sequential constraints.*

*Proof.* We establish the bound through the following chain of inequalities:

$$U(S_k^v) = \sum_{j=1}^{k} \Delta_{e_j} U(S_{j-1}^v) \qquad \text{(telescoping)}$$

$$\geq \sum_{j=1}^{k} (1-c)U(\{e_j\}) \qquad \text{(curvature: } \Delta_i U(S) \geq (1-c)U(\{i\}))$$

$$= (1-c) \sum_{i \in S_k^v} U(\{i\})$$

$$\geq (1-c) \sum_{i \in S_k^v} v(i) \qquad \text{(upper bound: } v(i) \leq U(\{i\}))$$

$$\geq (1-c) \sum_{i \in OPT_k^*} v(i) \qquad (S_k^v \text{ contains top-}k \text{ elements by } v)$$

$$\geq (1-c)^2 \sum_{i \in OPT_k^*} U(\{i\}) \qquad \text{(lower bound: } v(i) \geq (1-c)U(\{i\}))$$

$$\geq (1-c)^2 U(OPT_k^*) \qquad \text{(submodularity: } U(S) \leq \sum_{i \in S} U(\{i\}))$$

□

*Remark* G.5 (Interpretation of the $(1-c)^2$ Bound). The approximation ratio $(1-c)^2$ degrades quadratically with curvature $c$. When $c = 0$ (linear utility), we recover exact optimality. As $c \to 1$ (maximum diminishing returns), the guarantee weakens to 0. This explains why data valuation methods struggle on highly redundant datasets where points are nearly substitutable. Such datasets exhibit high curvature, causing the myopic linear approximation to deviate significantly from optimal selection.

This is a *worst-case* guarantee: it lower-bounds performance but does not predict typical performance. In practice, the bound is most useful for *explaining failures* (when methods perform poorly, high curvature provides a principled explanation) rather than for *predicting success*. Methods may substantially outperform the $(1-c)^2$ ratio on benign instances even when $c$ is moderate.

### G.4. Sequential Selection Guarantees

We now extend our analysis to the sequential selection setting, providing theoretical foundations for the results discussed in Section 4.1:

**Theorem G.6** (Sequential Selection Guarantee). *For any monotone submodular function $U$ with curvature $c$, let $\{S_t^v\}_{t=1}^{k}$ be a sequence constructed by selecting elements in descending order of their data values. Then: $\sum_{t=1}^{k} U(S_t^v) \geq (1-c)^2 \sum_{t=1}^{k} U(OPT_t^\pi)$ where $\{OPT_t^\pi\}_{t=1}^{k}$ represents the optimal sequence under sequential constraints (Definition G.1).*

*Proof.* We proceed by showing that the guarantee from Theorem G.4 extends to each prefix of the sequence. Let $S_t^v$ denote the set of first $t$ elements selected by our algorithm, and $OPT_t^\pi$ be an optimal set of size $t$ under sequential constraints.

From Theorem G.4, we know that for each $t \in [k]$:

$$U(S_t^v) \geq (1-c)^2 U(OPT_t^*) \geq (1-c)^2 U(OPT_t^\pi)$$

where the second inequality uses $U(OPT_t^*) \geq U(OPT_t^\pi)$ since unconstrained optimization can only improve over constrained optimization. Summing this inequality over all $t$ from 1 to $k$ directly yields:

$$\sum_{t=1}^{k} U(S_t^v) \geq (1-c)^2 \sum_{t=1}^{k} U(OPT_t^\pi)$$

This uniform approximation guarantee emerges from the fundamental properties of our selection process. The data values $v(\cdot)$ are computed once at the beginning and remain fixed throughout the selection process. As we proceed, at each step $t$, the set $S_t^v$ consists of the $t$ elements with the highest data values. Note that $OPT_t^\pi$ represents the optimal solution at step $t$ under the sequential constraint that elements must be chosen in a fixed order (nested constraint $S_1 \subset S_2 \subset \cdots \subset S_n$), distinguishing it from the unconstrained optimal solution $OPT_t^*$. $\square$

## H. End-to-end Approximation Guarantee for Bipartite Surrogate

### H.1. Submodularity of Coverage Utility

**Proposition H.1** (Submodularity of Coverage, Full Statement). *The coverage utility $\hat{U}(S)$ in Definition 5.1 is normalized ($\hat{U}(\emptyset) = 0$), monotone, and submodular. Moreover, if every training node has at least one validation neighbor (so that $\hat{U}(\{i\}) > 0$ for all $i$), then the curvature of $\hat{U}$ is well-defined and lies in $[0, 1]$. In general, the curvature can equal 1 due to redundancy (e.g., if a node's neighbors are entirely covered by others). A sufficient condition for $c < 1$ is that every training node has at least one* private *validation neighbor; more generally, $c < 1$ holds whenever no training node is fully redundant. Here curvature $c$ is defined with respect to the surrogate $\hat{U}$ (not the true utility $U$). Specifically, if each training node $i$ has at least one private validation neighbor $v$ such that $(i, v) \in E$ but $(j, v) \notin E$ for all $j \neq i$, then $\Delta_i \hat{U}(\mathcal{D} \setminus \{i\}) > 0$ and hence $c < 1$; otherwise fully redundant nodes can make $c = 1$.*

*Proof.* Let $N(S) \triangleq \{v \in X_{\text{valid}} : \exists u \in S, (u, v) \in E\}$ denote the validation neighborhood of $S$. Then $\hat{U}(S) = |N(S)|/|X_{\text{valid}}|$.

*Normalization* is immediate: $N(\emptyset) = \emptyset$, so $\hat{U}(\emptyset) = 0$.

*Monotonicity*: if $A \subseteq B$, then $N(A) \subseteq N(B)$, so $\hat{U}(A) \leq \hat{U}(B)$.

*Submodularity*: fix $A \subseteq B \subseteq X_{\text{train}}$ and $i \notin B$. The marginal gain at $A$ is

$$\hat{U}(A \cup \{i\}) - \hat{U}(A) = \frac{|N(\{i\}) \setminus N(A)|}{|X_{\text{valid}}|},$$

and analogously at $B$. Since $N(A) \subseteq N(B)$, we have $N(\{i\}) \setminus N(A) \supseteq N(\{i\}) \setminus N(B)$, so

$$\hat{U}(A \cup \{i\}) - \hat{U}(A) \geq \hat{U}(B \cup \{i\}) - \hat{U}(B).$$

This is the diminishing-marginal-returns characterization of submodularity.

*Curvature.* If every training node $i$ has at least one validation neighbor, $\hat{U}(\{i\}) > 0$, so curvature is well-defined and lies in $[0, 1]$ by monotonicity and submodularity. The terminal marginal is

$$\Delta_i \hat{U}(X_{\text{train}} \setminus \{i\}) = \frac{|N(\{i\}) \setminus N(X_{\text{train}} \setminus \{i\})|}{|X_{\text{valid}}|},$$

which counts the validation neighbors of $i$ that are *not* covered by any other training node. If every training node has at least one such private neighbor, this marginal is positive for every $i$, so $c < 1$. Conversely, if some training node's entire neighborhood is covered by others, the corresponding marginal is zero, giving $c = 1$. $\square$

### H.2. End-to-end Approximation Theorem

This section provides the complete formulation and proof for the theoretical guarantee of our bipartite graph-based approach.

**Theorem H.2** (End-to-end Approximation via a Submodular Surrogate). *Fix a cardinality budget $k$ and let $\epsilon \in [0, 1)$. Let $\hat{U}$ be a normalized, monotone, submodular surrogate utility (e.g., coverage), and let $U : 2^{\mathcal{D}} \to \mathbb{R}_{\geq 0}$ be the true utility. (**Non-negativity of $U$ is required for the multiplicative approximation ratio to be meaningful.**) Assume the following uniform "sandwich" conditions hold for all $S \subseteq \mathcal{D}$:*

*(i) $\hat{U}(S) \geq (1 - \epsilon) U(S)$*                    *(surrogate lower-bounds $U$ up to $1 - \epsilon$)*
*(ii) $\hat{U}(S) \leq U(S)$*                        *(U upper-bounds the surrogate; no over-estimation)*

*Let $G$ be the size-$k$ greedy solution maximizing $\hat{U}$, and let $O^* \in \arg\max_{|S|=k} U(S)$ be an optimal size-$k$ solution for the true utility. Then*

$$U(G) \geq (1 - \epsilon)(1 - 1/e) U(O^*).$$

*Note that for monotone $U$, $\arg\max_{|S|=k} U(S)$ and $\arg\max_{|S| \leq k} U(S)$ yield the same optimum value since the optimal solution will always use the full budget $k$; we state the theorem with $|S| = k$ for consistency.*

*Proof.* Let $G = \{g_1, \ldots, g_k\}$ be the greedy solution on $\hat{U}$ and $O^* = \arg\max_{|S|=k} U(S)$.

By the classical result for submodular maximization, greedy on $\hat{U}$ satisfies:

$$\hat{U}(G) \geq (1 - 1/e)\hat{U}(O^*_{\hat{U}})$$

where $O^*_{\hat{U}} = \arg\max_{|S|=k} \hat{U}(S)$.

Since $O^*_{\hat{U}}$ maximizes $\hat{U}$ over $|S| = k$, we have $\hat{U}(O^*_{\hat{U}}) \geq \hat{U}(O^*)$. Combined with the sandwich conditions:

$$
\begin{aligned}
U(G) &\geq \hat{U}(G) & \text{(by condition (ii): } \hat{U} \leq U) \\
&\geq (1 - 1/e)\hat{U}(O^*_{\hat{U}}) & \text{(greedy guarantee)} \\
&\geq (1 - 1/e)\hat{U}(O^*) & \text{(optimality of } O^*_{\hat{U}} \text{ for } \hat{U}) \\
&\geq (1 - 1/e)(1 - \epsilon)U(O^*) & \text{(by condition (i): } \hat{U} \geq (1 - \epsilon)U)
\end{aligned}
$$

Therefore $U(G) \geq (1 - \epsilon)(1 - 1/e)U(O^*)$.      $\square$

*Remark* H.3 (On the Conditions of Theorem H.2). The sandwich conditions (i) and (ii) are sufficient (not necessary) assumptions that make the transfer of the greedy $(1 - 1/e)$ guarantee transparent; in general they may fail for coverage surrogates, so we treat them as an idealized regime and empirically assess over-estimation/under-estimation on held-out sampled subsets.

- **Condition (ii) ($\hat{U}(S) \leq U(S)$).** For coverage-based surrogates, this inequality is not guaranteed in general. Coverage can be optimistic when the threshold is loose. In practice, we treat (ii) as a modeling desideratum and empirically monitor whether $\hat{U}$ tends to overestimate $U$ on held-out sampled subsets.
- **Condition (i) ($\hat{U}(S) \geq (1 - \epsilon)U(S)$).** This is a uniform multiplicative lower bound, which is typically too strong to infer from average regression error alone. Instead of interpreting $\epsilon$ as a theoretical constant implied by MSE, we report empirical approximation quality using sampled subsets and metrics such as MAE and MSE (Table 3).

Overall, while Theorem H.2 provides a clean sufficient condition for end-to-end guarantees, our empirical results primarily support the practical claim that the bipartite coverage surrogate is a significantly better predictor of the true utility than linear baselines, which in turn leads to substantially improved greedy selection.

# I. Computational Complexity Analysis

We present a detailed analysis of the computational complexity for our bipartite graph-based approach described in Algorithm 1. Let $n = |X_{\text{train}}|$ denote the number of training points, $m = |X_{\text{valid}}|$ the number of validation points, and $d$ the feature dimensionality. The computational cost comprises three main components.

The first component involves computing pairwise distances between training and validation points, requiring $\mathcal{O}(nmd)$ operations when using Euclidean distance. While this step dominates the computation for high-dimensional data, it can be optimized through vectorized BLAS operations or approximate nearest-neighbor search techniques.

The second component performs threshold search across $N_\tau$ candidate values. For each threshold, the algorithm evaluates prediction error on $K$ random subsets through two operations: constructing edges by scanning the distance matrix ($\mathcal{O}(N_\tau nm)$ comparisons) and computing coverage for $K$ subsets of average size $k$ ($\mathcal{O}(N_\tau K(k+m))$ operations using efficient bit-set implementations). Since $k \ll m$ and both $K$ and $N_\tau$ are small constants in practice (typically $K = 100$, $N_\tau = 30$), the edge construction dominates this phase.

The final component executes greedy selection after determining the optimal threshold $\tau^*$. Using a priority queue with lazy updates to track residual node degrees, this phase requires $\mathcal{O}(n \log n + |E_{\tau^*}|)$ operations, where $|E_{\tau^*}| \leq nm$ represents the number of edges in the sparsified graph.

The total computational complexity thus becomes:

$$\mathcal{O}(nmd) + \mathcal{O}(N_\tau nm) + \mathcal{O}(n \log n + |E_{\tau^*}|)$$

Since $d$ and $N_\tau$ are dataset-specific constants, the algorithm scales linearly with $nm$. The space complexity is $\mathcal{O}(nm)$, primarily storing the distance matrix and edge set.

In practical data valuation scenarios where $m \ll n$, our method offers significant computational advantages over traditional Shapley-based approaches that require extensive model retraining (Ghorbani & Zou, 2019; Wang & Jia, 2023). Furthermore, both distance computation and edge construction phases are naturally parallelizable and GPU-compatible, enabling efficient scaling to larger datasets as demonstrated in our experimental results.

## J. Experimental Details

We provide detailed descriptions of our experimental setup, including baseline methods, datasets, and evaluation protocols.

### J.1. Target Model

In our experiments, the target model used for data valuation is the Logistic Regression model from the Scikit-learn machine learning library. The data valuation process involves repeatedly training this Logistic Regression model on different data subsets to evaluate the marginal contributions of each training sample to the model's test performance, thereby deriving the corresponding data value scores.

### J.2. Baseline Methods

We evaluate our approach against the following state-of-the-art data valuation methods:

- `Random`: A naive baseline that randomly selects data points

- `LOO` (Leave-one-out): Measures each point's marginal contribution by removing it from the full dataset

- `Influence Function` (Koh & Liang, 2017): Approximates data influence using model gradients

- `Data Shapley` (Ghorbani & Zou, 2019): Assigns values based on average marginal contributions across different subset sizes

- `Beta Shapley` (Kwon & Zou, 2022): Generalizes Shapley value by relaxing the efficiency axiom

- `Data Banzhaf` (Wang & Jia, 2023): Captures influence through binary marginal contributions

- `AME` (Average Marginal Effect) (Lin et al., 2022): Estimates influence using sparse regression

- `DVRL` (Data Valuation using Reinforcement Learning) (Yoon et al., 2020): Learns data values through policy optimization

- `DataOob` (Data Out-of-Bag) (Kwon & Zou, 2023): Uses out-of-bag estimates from bagging models to evaluate data utility

All baseline methods are implemented using OpenDataVal (Jiang et al., 2023), ensuring standardized implementation and fair comparison. All experiments were conducted on CPU using Scikit-learn's built-in multi-threading capabilities for parallel acceleration.

## J.3. Dataset Details

Our experiments use eight diverse datasets from OpenML (Feurer et al., 2021), spanning tabular, text, and image domains:

- **2dplanes** (ID 727): A synthetic dataset for binary classification

- **nomao** (ID 1486) (Candillier & Lemaire, 2012): A real-world dataset for duplicate detection

- **bbc-embeddings** (Greene & Cunningham, 2006): Text classification dataset with news articles

- **MiniBooNE** (ID 43974) (Roe et al., 2005): Particle physics dataset for binary classification

- **digits** (Xu et al., 1992): Handwritten digit recognition dataset

- **election** (MIT Election Data and Science Lab, 2017): Election outcome prediction dataset

- **electricity** (ID 44080) (Gama et al., 2004): Time series dataset for price movement prediction

- **fried** (ID 901): Synthetic tabular/categorical dataset

These datasets are commonly used in data valuation research (Yoon et al., 2020; Ghorbani & Zou, 2019; Kwon & Zou, 2023) and represent a diverse range of learning tasks and data characteristics.

## J.4. Evaluation Protocol Details

For each dataset, we follow a rigorous evaluation protocol:

1. **Data Splitting**: Each dataset $\mathcal{D}$ is randomly split into:
   - Training set $\mathcal{D}_{train}$
   - Validation set $\mathcal{D}_{valid}$ (for computing utility functions)
   - Test set $\mathcal{D}_{test}$ (for evaluation)

2. **Value Assignment**: Data valuation methods compute values for each training point using validation set performance

3. **Sequential Selection**:
   - Rank training points by assigned values in descending order
   - Iteratively add points following this ranking
   - Train model on selected subset at each step
   - Record test accuracy

4. **Performance Curves**: Plot test accuracy versus selection size $k \in [1, |\mathcal{D}_{train}|]$

5. **Experimental Control**:
   - All methods limited to 1000 model retraining steps
   - Results averaged over 20 independent runs
   - Consistent random seeds used across methods

This protocol follows standard practices in data valuation literature (Ghorbani & Zou, 2019; Kwon & Zou, 2022; Tarun et al., 2024) and ensures fair and comprehensive evaluation of different methods.

## J.5. RQ1 Experimental Settings

To establish performance bounds and quantify the limitations of existing data valuation methods, we conduct experiments with the following specific settings:

- **Data Sampling**: For each dataset:
    - Training set: 20 points
    - Validation set: 100 points
    - Test set: 500 points

- **Random Seeds**: 20 independent trials using seeds from 10 to 200 with step 10

- **Methods Compared**:
    - Optimal strategy: `DynamicProgramming` (Algorithm 2)
    - Existing methods: `Random`, `LOO`, `Influence`, `DataShap`, `BetaShap`, `Banzhaf`, `AME`, `DVRL`, `DataOob`

- **Evaluation**: Test accuracy evaluated by sequentially adding points based on their assigned values

The selection curves are generated by plotting test accuracy against selection size $k$, with results averaged over all trials.

## J.6. RQ3 Experimental Settings

We evaluate our bipartite graph-based approximation approach `Bipartite` with the following settings:

- **Dataset Splitting**:
    - Standard datasets:
        * Training set: 50 points
        * Validation set: 50 points
        * Test set: 500 points
    - Large datasets (digits and bbc-embeddings):
        * Training set: 100 points
        * Validation set: 100 points
        * Test set: 1000 points

- **Evaluation Protocol**:
    - Model retraining steps: 1000
    - Independent runs: 20
    - Metric: Test accuracy vs. selection size

**Budget-500 protocol.** The split above is used for the standard RQ3 selection curves and the detailed small-budget selection table. The large-scale Budget-500 table in the main text uses a separate protocol with 500 training candidates for every dataset. Standard datasets use 50 validation and 500 test points, while digits and bbc-embeddings use 100 validation and 1000 test points. The reported Budget-500 value is the mean test accuracy over the top-$k$ selection curve for $k = 1, \ldots, 500$, rather than only the endpoint at $k = 500$.

**DVRL training budget calibration.** Following OpenDataVal's default, we use `DVRL(rl_epochs=1000, rl_batch_size=32)`, which makes `pred_model.fit()` approximately 1000 times in the RL loop – matching the 1000-retraining budget of the other sampling-based methods (`InfluenceSubsample`, `DataBanzhaf`, `DataOob`, `BipartiteMatchingEvaluator`, and `AME` via four bagging proportions of 250). Each DVRL `pred_model.fit()` call uses a minibatch of 32 examples, intrinsic to its REINFORCE-style training.

## J.7. LLM Fine-Tuning Data Selection (DATE-LM)

To assess whether our selection framework transfers to large-scale LLM fine-tuning, we evaluate on DATE-LM (Jiao et al., 2025). We refer to our adapted method as `BipCov` (Bipartite Coverage), which applies the core bipartite graph coverage framework from Section 5 with RAG-style embeddings suitable for instruction selection.

### J.7.1. EXPERIMENTAL SETUP

**Task and Protocol.** DATE-LM (Table 3) studies single-task instruction fine-tuning: given a target evaluation task, methods select a training subset from a large instruction pool to maximize downstream task performance. We follow the official DATE-LM Table 3 protocol for direct comparison with published baselines.

**Training Pool.** Following DATE-LM, we sample 200,000 instructions from Tulu 3 unfiltered (Lambert et al., 2025) using seed 42. For multi-turn conversations we keep only the first user–assistant turn, consistent with the DATE-LM data pipeline. The pool contains diverse instruction-response pairs spanning general knowledge, reasoning, coding, and creative tasks.[1]

**Reference Set.** For each target task, we sample 100 prompt+label examples from the evaluation dataset using seed 42.[2] These reference examples guide task-aware data selection.

**Selection Budget.** Each method produces a score vector over the 200k pool and selects the top 10,000 examples (5%) for fine-tuning.

**Base Model and Fine-Tuning.** We fine-tune `meta-llama/Llama-3.1-8B` (base, not instruction-tuned) using LoRA (Hu et al., 2021) with the following hyperparameters:

- LoRA rank: 128, LoRA alpha: 512, LoRA dropout: 0.1
- Target modules: `q_proj`, `k_proj`, `v_proj`, `o_proj`[3]
- Learning rate: $2 \times 10^{-5}$ with linear warmup for the first 3% of training steps, followed by cosine annealing to 0
- Effective batch size: 128 (micro batch 1 $\times$ gradient accumulation 128, single device)
- Training epochs: 2 (approximately 156 optimization steps)
- Max sequence length: 2048 tokens (truncated)
- Precision: bf16 (full bf16, no mixed precision)
- Gradient clipping: none
- Loss masking: assistant-only (instruction tokens are masked via labels $= -100$)

**Evaluation Tasks.** We evaluate on three diverse benchmarks using official DATE-LM evaluation scripts:

- **MMLU** (Hendrycks et al., 2021): 14,042 questions across 57 subjects, evaluated with 0-shot accuracy.
- **GSM8K** (Cobbe et al., 2021): 1,319 grade-school math problems, evaluated with 8-shot chain-of-thought exact match.
- **BBH** (Suzgun et al., 2022): 6,511 questions across 27 challenging BIG-Bench subtasks, evaluated with 3-shot exact match.

### J.7.2. BIPCOV IMPLEMENTATION DETAILS

**Adapting BipCov to DATE-LM.** DATE-LM exposes data selection methods through a per-example metric vector (`metrics.npy`) over the pool; the pipeline selects top-$k$ by this score. To instantiate BipCov, we build a bipartite graph between pool instructions and the task reference set using cosine similarity in an embedding space.

---

[1]The official DATE-LM data module additionally holds out a 10% validation split (22,222 examples) for logging val loss during training; this split is not used for selection, model selection, or early stopping.

[2]DATE-LM's Table 3 protocol constructs the reference set using prompt+label examples; we follow this setting for apples-to-apples comparison with DATE-LM baselines. We note that including labels in the reference set introduces potential label leakage, which is an inherent property of the DATE-LM benchmark design rather than our method.

[3]Implemented as `lora_query=lora_key=lora_value=lora_projection=True` in the LitGPT/DATE-LM trainer; `lora_projection` is LitGPT's name for `o_proj`. MLP and head layers are not adapted.

**Graph Construction.** We sparsify the bipartite graph by connecting each reference example to its top-$L$ nearest pool examples. We fix $L = 200$ throughout all experiments, which provides sufficient coverage while keeping the graph sparse. This hyperparameter was chosen based on preliminary experiments showing stable performance for $L \in [100, 500]$.

**Greedy Selection.** We apply lazy greedy maximization of the maximum-coverage objective to generate a selection order up to $k = 10{,}000$. The algorithm maintains a set of covered reference examples and iteratively selects the pool example that covers the most uncovered references. When binary coverage saturates (i.e., all reference examples are covered before reaching budget $k$), we fill the remaining budget by ranking uncovered pool examples by their mean cosine similarity to the reference set. This fallback ensures a complete top-$k$ ordering compatible with the DATE-LM pipeline while maintaining the coverage-first priority. In the ref-aligned Llama-3.1-8B last-token hidden-state score files used for Table 6, binary coverage saturated after selecting 2, 1, and 5 examples for MMLU, GSM8K, and BBH, respectively; the remaining positions up to 10,000 were filled by the mean-similarity fallback. Thus the DATE-LM results should be interpreted as coverage-prioritized, reference-aligned similarity selection rather than pure coverage across the entire 10k budget.

**RAG-style Embeddings.** For computing bipartite edge weights, we explore both LLM internal representations and external retrieval encoders:

- **LLM hidden states**: Last-token or position-weighted mean pooled hidden states from `Llama-3.1-8B`, as used by Rep-Sim and RDS+ (4096 dimensions).
- **Retrieval encoders**: Dense retrieval models including BGE-large-v1.5 (1024 dim), E5-large-v2 (1024 dim), Qwen3-Emb-8B, GTE-Qwen2-7B, NV-Embed-v2, and GritLM-7B.

We treat each pool instruction as a "document" and each reference prompt as a "query", applying model-specific formatting (e.g., `query:`/`passage:` prefixes for E5) when applicable, analogous to retrieval in RAG pipelines.

### J.7.3. BASELINE METHODS

We compare against the following DATE-LM baselines:

- **Random**: Uniform random selection from the pool (DATE-LM reports average over 3 random seeds).
- **BM25** (Trotman et al., 2014): Lexical matching between pool instructions and reference prompts.
- **Rep-Sim**: Cosine similarity using *last-token* hidden states from Llama-3.1-8B; each pool example is scored by its average similarity to reference examples.
- **RDS+** (Ivison et al., 2025): Similar to Rep-Sim but uses *position-weighted mean pooling* over all token hidden states: $\mathbf{e} = \sum_{i=1}^{L} w_i \mathbf{h}_i$ where $w_i = i / \sum_{j=1}^{L} j$.
- **Grad-Sim**: Gradient similarity between pool examples and reference examples using a warmup LoRA checkpoint.
- **LESS** (Xia et al., 2024b): Learned embedding-based scoring using gradient features from a warmup checkpoint.

### J.7.4. COMPUTATIONAL COST

Table 5 summarizes the computational requirements recorded from the synced DATE-LM GPU run logs. The dominant cost is embedding computation and downstream fine-tuning/evaluation; BipCov's coverage-based scoring itself takes only seconds on CPU once embeddings are available.

*Table 5.* Measured computational cost from the synced DATE-LM run records. Timing rows below are from the $4{\times}$H100 NVL single-seed Table 7 and embedding-ablation runs, where each train/eval job occupies one GPU. The multi-seed headline results in Table 6 were run on $1{\times}$H200; those records preserve hardware, commands, and raw metrics, but not complete per-stage wall-clock timing.

| Stage | Recorded time | Notes |
|---|---|---|
| BGE train embeddings (200k pool) | 35.9 min | One-time; shared across tasks |
| E5 train embeddings (200k pool) | 35.4 min | One-time; shared across tasks |
| Llama hidden-state embeddings (200k pool) | 2.8–3.0 h | Shared by RDS+, Rep-Sim, BipCov |
| BipCov scoring/selection | 0–7 s per task | CPU scoring after embeddings |
| MMLU train+official eval | 59–73 min | LoRA train, convert, and evaluate |
| GSM8K train+official eval | 73–79 min | LoRA train, convert, and evaluate |
| BBH train+official eval | 162–183 min | LoRA train, convert, and evaluate |
| **One method, 3 tasks after embeddings** | 4.95–5.58 GPU-h | Occupied single-GPU job time |

### J.7.5. WHY COVERAGE HELPS

Compared with average-similarity scoring (Rep-Sim, RDS+), BipCov encourages early diversity with respect to the reference distribution: each selected example is rewarded for covering previously uncovered reference prompts, reducing redundancy while maintaining task alignment via the similarity graph. Once all reference prompts are covered, the remaining budget is filled by the mean-similarity fallback described above.

Consider a scenario where the reference set contains questions about geography, mathematics, and history. Average-similarity methods may over-select examples similar to the most frequent reference type, while BipCov prioritizes balanced coverage across reference topics before adding additional high-similarity examples.

The embedding-backend ablations (Table 8) further suggest that stronger retrieval encoders provide more faithful semantic neighborhoods, which can translate into better downstream fine-tuning performance. Notably, Qwen3-Emb-8B achieves the highest average score (63.71%), suggesting that larger, task-aware embedding models may provide better guidance for instruction selection.

### J.7.6. RESULTS AND ANALYSIS

Table 6 reports mean $\pm$ std over three fine-tuning seeds (42, 1337, 2025) for `Random`, RDS+, and our `BipCov` selection. The `BipCov` row uses the ref-aligned prompt+label reference construction with Llama-3.1-8B last-token hidden-state embeddings; the weighted-mean LLM embedding variant appears separately as an embedding ablation in Table 8. BipCov achieves the best average performance (63.89%) with a +1.01% improvement over RDS+ (62.88%), demonstrating consistent gains across different training seeds.

For completeness, we also reproduce the broader DATE-LM Table 3 baseline set under the same official pipeline (single fine-tuning seed 1337; Table 7). BipCov with BGE embeddings achieves the best average (63.56%), outperforming all reproduced baselines including gradient-based methods (Grad-Sim, LESS) that require additional warmup training.

Table 8 reports representation (embedding) ablations for BipCov. Key observations:

- **Retrieval encoders outperform LLM hidden states**: BGE (63.56%) and Qwen3-Emb-8B (63.71%) outperform weighted-mean LLM embeddings (62.89%), suggesting that models trained for semantic retrieval provide better similarity signals for coverage-based selection.
- **Task-specific trade-offs**: Different embeddings excel on different tasks. BGE achieves the best BBH score (67.14%), while Qwen3-Emb-8B leads on MMLU (62.50%) and GSM8K (63.00%). This suggests potential for task-adaptive embedding selection.
- **Embedding dimension is not the bottleneck**: GritLM-7B (7B parameters) underperforms BGE-large (335M parameters), indicating that embedding quality matters more than model size for this application.

*Table 6.* LLM fine-tuning data selection on DATE-LM (Jiao et al., 2025). Results are mean ($\pm$ std) in % over three fine-tuning seeds (42, 1337, 2025) using the official DATE-LM pipeline on `MMLU` (acc), `GSM8K` (EM), and `BBH` (EM). `BipCov` (ref-aligned) uses Llama-3.1-8B last-token hidden-state embeddings with prompt+label reference examples matching the DATE-LM protocol; Table 8 reports weighted-mean LLM embeddings as a separate ablation. Best results are shown in **bold**.

| Method | MMLU | GSM8K | BBH | Avg |
|---|---|---|---|---|
| Random | 60.00 | 61.89 | 66.58 | 62.82 |
| | ($\pm$0.17) | ($\pm$0.58) | ($\pm$0.30) | ($\pm$0.31) |
| RDS+ | 59.58 | 62.24 | **66.80** | 62.88 |
| | ($\pm$0.07) | ($\pm$0.73) | ($\pm$0.21) | ($\pm$0.25) |
| BipCov (ref-aligned) | **61.87** | **63.53** | 66.27 | **63.89** |
| | ($\pm$0.14) | ($\pm$0.41) | ($\pm$0.58) | ($\pm$0.34) |

## K. RQ2 Experimental Validation of Curvature Impact

To empirically examine the theoretical guarantees established in Theorem 4.4 and the limitations described in Remark G.5, we conduct controlled experiments that systematically vary data point substitutability as a qualitative proxy for utility curvature. Our experimental design uses message passing mechanisms to control the degree of within-class feature similarity, which affects the substitutability structure of the learned utility.

*Table 7.* Reproduced DATE-LM Table 3 baselines (single training seed 1337) using the official DATE-LM pipeline. Values are in % (higher is better). Best results are shown in **bold**.

| Method | MMLU | GSM8K | BBH | Avg |
|---|---|---|---|---|
| Random Avg | 60.39 | 59.64 | 66.40 | 62.14 |
| BM25 | 59.85 | 58.98 | 62.63 | 60.49 |
| RepSim | 61.42 | 58.45 | 66.20 | 62.03 |
| RDS+ | 60.63 | 61.41 | 66.23 | 62.75 |
| Grad Sim | **62.07** | 56.71 | 64.44 | 61.07 |
| LESS | 61.10 | 57.77 | 64.68 | 61.18 |
| BipCov (BGE emb) | 61.81 | **61.71** | **67.14** | **63.56** |

*Table 8.* Representation (embedding) ablations for BipCov under the DATE-LM Table 3 pipeline (single training seed 1337). Only the embedding backend changes; the BipCov objective and greedy selection hyperparameters are fixed. Values are in % (higher is better). Best results are shown in **bold**.

| BipCov variant | MMLU | GSM8K | BBH | Avg |
|---|---|---|---|---|
| BipCov (weighted-mean emb) | 60.49 | 61.87 | 66.32 | 62.89 |
| BipCov (BGE emb) | 61.81 | 61.71 | **67.14** | 63.56 |
| BipCov (E5 emb) | 62.14 | 60.58 | 65.50 | 62.74 |
| BipCov (Qwen3-Emb-8B) | **62.50** | **63.00** | 65.62 | **63.71** |
| BipCov (GTE-Qwen2-7B) | 61.47 | 62.55 | 66.40 | 63.47 |
| BipCov (NV-Embed-v2) | 62.13 | 59.21 | 66.14 | 62.49 |
| BipCov (GritLM-7B) | 61.07 | 61.11 | 65.56 | 62.58 |

Our experimental setup consists of a three-class classification problem where we employ a graph-based message passing scheme that updates each point's features by aggregating information from its within-class neighbors. The aggregation proportion increases from $0.0$ to $1.0$, systematically increasing substitutability, our proxy for curvature. When the proportion is small, points maintain distinctive features, corresponding to low substitutability. As the proportion increases, points within each class become increasingly similar through feature averaging, leading to high substitutability.

The experimental results are consistent with our theoretical analysis in three aspects. First, under low substitutability (proportion $\leq 0.3$), all methods achieve strong performance with BetaShapley obtaining a mean accuracy of $0.706$, Banzhaf ($0.720$) and Shapley ($0.738$). This aligns with the $(1-c)^2$ approximation guarantee from Theorem 4.4 when the curvature proxy is small. Second, as substitutability increases (proportion $\geq 0.5$), we observe significant performance degradation across all methods, with mean accuracies dropping to approximately $0.59$ at proportion $1.0$. This deterioration is consistent with the quadratic decay in our theoretical bound as curvature approaches $1$. Third, the convergence in performance between different methods at high substitutability is consistent with our analysis that all game-theoretic approaches face similar fundamental limitations under strong substitution effects.

These results are **consistent with** Theorem 4.4's prediction that higher substitutability (which we use as a curvature proxy) worsens the performance of marginal-average, myopic value rankings, though we do not claim to directly estimate the true curvature $c$ of the learned utility.

## L. Algorithm for Computing Sequential Optimal Data Values

We present the complete algorithm for computing optimal data values through dynamic programming in Algorithm 2. The algorithm proceeds in two phases: (1) backward induction to compute the optimal value function and policy, and (2) forward traversal to derive data values from the optimal selection sequence. The algorithm's first phase implements backward dynamic programming, computing the optimal value function $V[s]$ and policy $\sigma^*[s]$ for each state $s$. The second phase constructs the optimal selection sequence and assigns values inversely proportional to selection order. While this algorithm yields optimal values, its computational complexity is exponential in the dataset size due to the need to evaluate all possible subset states.

*Convention.* The array $V[\cdot]$ in Algorithm 2 (square brackets) stores a shifted form of the value-to-go $V(\cdot)$ in Equation (2) (parentheses): its backward recursion maintains $V[s] = U(s) + V(s)$, folding the current prefix utility $U(s)$ into the stored value, with boundary $V[\mathcal{D}] = U(\mathcal{D})$. This is purely a bookkeeping choice for a compact recursion: because

$V[s \cup \{a\}] = U(s \cup \{a\}) + V(s \cup \{a\})$ is exactly the quantity maximized inside Equation (2), the selected action $\sigma^*[s] = \arg\max_{a \in \mathcal{D} \setminus s} V[s \cup \{a\}]$ coincides with the optimal policy $\arg\max_a \{U(s \cup \{a\}) + V(s \cup \{a\})\}$, so Algorithm 2 yields exactly the same optimal permutation $\pi^*$ and data values $v^*$.

---

**Algorithm 2** Computing Optimal Data Values for Selection

---

1: **Input:** dataset $\mathcal{D}$, utility function $U : 2^{\mathcal{D}} \to \mathbb{R}$
2: *// Initialize value for full dataset*
3: Initialize $V[\mathcal{D}] \leftarrow U(\mathcal{D})$
4: *// Backward phase: compute optimal value function*
5: **for** $t = |\mathcal{D}| - 1$ **to** $0$ **do**
6:    **for** each state $s \subseteq \mathcal{D}$ with $|s| = t$ **do**
7:       $V[s] \leftarrow U(s) + \max_{a \in \mathcal{D} \setminus s} V[s \cup \{a\}]$
8:       $\sigma^*[s] \leftarrow \mathrm{argmax}_{a \in \mathcal{D} \setminus s} V[s \cup \{a\}]$
9:    **end for**
10: **end for**
11: *// Forward phase: derive data values*
12: Initialize $s \leftarrow \emptyset$
13: **for** $t = 1$ **to** $|\mathcal{D}|$ **do**
14:    $a_t \leftarrow \sigma^*[s]$ *// Select next element*
15:    $v^*[a_t] \leftarrow |\mathcal{D}| - t$ *// Assign value*
16:    $s \leftarrow s \cup \{a_t\}$
17: **end for**
18: **Return:** value function $v^*$

---

## M. Detailed Analysis of RQ1 Results

Building upon the performance gaps identified in Section 7.1, here we provide a detailed analysis of the selection curves to understand how existing methods compare with optimal sequential selection across different data budgets. Several key findings emerge:

First, the optimal selection `DynamicProgramming` consistently outperforms all existing methods across datasets, with particularly pronounced gaps in early selection stages ($k \leq 5$). This indicates that current approaches significantly underperform in identifying the most valuable initial samples. Second, the performance gap varies notably across datasets, suggesting dataset-specific challenges. The gap is particularly prominent in structured datasets like bbc-embeddings and nomao, where optimal selection demonstrates steep initial performance improvements that existing methods fail to match. Third, existing methods exhibit varying patterns of suboptimality. Game-theoretic approaches (`DataShap`, `BetaShap`, `Banzhaf`) tend to perform similarly to each other but consistently fall short of `DynamicProgramming`. Learning-based methods (`DVRL`, `AME`) and influence-based approaches show competitive performance in some cases but lack consistency across datasets. Notably, all methods cannot match the optimal strategy's ability to achieve both rapid initial improvement and sustained performance gains.

These detailed observations complement the quantitative performance gaps reported in Section 7.1, providing deeper insights into how different data valuation approaches behave across varying selection budgets.

## N. Detailed Analysis of Data Selection Performance

To quantify the selection performance improvements achieved by our bipartite graph-based approach `Bipartite`, we present detailed numerical results comparing against baseline methods. For each dataset and method, we compute the mean test accuracy and standard deviation across 20 independent runs. The experimental protocol follows Section 7.3.

The results reinforce our observations from the selection curves in Section 7.3. Our method `Bipartite` achieves the highest average accuracy, ranking first on seven of the eight datasets, with particularly substantial improvements on complex datasets like bbc-embeddings (0.878 accuracy, compared to 0.809 for the best baseline) and digits (0.764 accuracy versus 0.680 for the next best method). Even on datasets where baseline methods perform relatively well, such as nomao and MiniBooNE, our approach `Bipartite` still maintains a clear performance advantage while demonstrating more stable

*Table 9.* Mean test accuracy ($\pm$ standard deviation) of different data selection methods across 20 independent runs. Best mean results for each dataset are shown in **bold**.

| Dataset | Random | LOO | Influence | DataShap | BetaShap | Banzhaf | AME | DVRL | DataOob | Bipartite |
|---|---|---|---|---|---|---|---|---|---|---|
| 2dplanes | 0.723 | 0.720 | 0.729 | 0.740 | **0.747** | 0.729 | 0.724 | 0.728 | 0.734 | 0.745 |
| | ($\pm$0.098) | ($\pm$0.103) | ($\pm$0.109) | ($\pm$0.106) | ($\pm$0.094) | ($\pm$0.113) | ($\pm$0.117) | ($\pm$0.105) | ($\pm$0.090) | ($\pm$0.069) |
| nomao | 0.812 | 0.806 | 0.788 | 0.815 | 0.815 | 0.782 | 0.778 | 0.777 | 0.682 | **0.852** |
| | ($\pm$0.148) | ($\pm$0.171) | ($\pm$0.189) | ($\pm$0.163) | ($\pm$0.158) | ($\pm$0.194) | ($\pm$0.203) | ($\pm$0.190) | ($\pm$0.252) | ($\pm$0.088) |
| bbc-embeddings | 0.805 | 0.803 | 0.743 | 0.790 | 0.809 | 0.711 | 0.803 | 0.653 | 0.561 | **0.878** |
| | ($\pm$0.228) | ($\pm$0.246) | ($\pm$0.284) | ($\pm$0.261) | ($\pm$0.240) | ($\pm$0.312) | ($\pm$0.251) | ($\pm$0.327) | ($\pm$0.337) | ($\pm$0.146) |
| MiniBooNE | 0.722 | 0.703 | 0.678 | 0.711 | 0.731 | 0.679 | 0.691 | 0.689 | 0.726 | **0.754** |
| | ($\pm$0.092) | ($\pm$0.100) | ($\pm$0.107) | ($\pm$0.093) | ($\pm$0.086) | ($\pm$0.109) | ($\pm$0.108) | ($\pm$0.119) | ($\pm$0.077) | ($\pm$0.066) |
| digits | 0.619 | 0.680 | 0.646 | 0.649 | 0.647 | 0.610 | 0.662 | 0.422 | 0.338 | **0.764** |
| | ($\pm$0.342) | ($\pm$0.318) | ($\pm$0.346) | ($\pm$0.335) | ($\pm$0.332) | ($\pm$0.361) | ($\pm$0.324) | ($\pm$0.367) | ($\pm$0.349) | ($\pm$0.245) |
| election | 0.566 | 0.537 | 0.555 | 0.580 | 0.578 | 0.539 | 0.586 | 0.565 | 0.263 | **0.601** |
| | ($\pm$0.224) | ($\pm$0.245) | ($\pm$0.233) | ($\pm$0.200) | ($\pm$0.198) | ($\pm$0.237) | ($\pm$0.202) | ($\pm$0.214) | ($\pm$0.228) | ($\pm$0.182) |
| electricity | 0.639 | 0.627 | 0.623 | 0.650 | 0.656 | 0.628 | 0.632 | 0.637 | 0.638 | **0.669** |
| | ($\pm$0.082) | ($\pm$0.081) | ($\pm$0.085) | ($\pm$0.078) | ($\pm$0.065) | ($\pm$0.087) | ($\pm$0.086) | ($\pm$0.081) | ($\pm$0.072) | ($\pm$0.063) |
| fried | 0.702 | 0.698 | 0.700 | 0.710 | 0.719 | 0.707 | 0.702 | 0.730 | 0.719 | **0.741** |
| | ($\pm$0.087) | ($\pm$0.099) | ($\pm$0.108) | ($\pm$0.103) | ($\pm$0.095) | ($\pm$0.106) | ($\pm$0.109) | ($\pm$0.084) | ($\pm$0.085) | ($\pm$0.065) |
| Average | 0.698 | 0.697 | 0.683 | 0.706 | 0.713 | 0.673 | 0.697 | 0.650 | 0.583 | **0.750** |

behavior across different runs. These comprehensive results validate our theoretical analysis that the bipartite graph approximation effectively preserves the essential structure of the utility function while enabling efficient computation.

# O. Future Work

We outline several promising research directions that could extend our framework.

## O.1. Non-myopic Planning with Expressive Models

While our current work focuses on myopic planning with structured bipartite models, an interesting direction would be exploring non-myopic planning strategies with more expressive surrogate models like neural networks, which have already been adopted in data utility learning (Wang et al., 2021). Such models might better approximate the ground truth utility but lack the theoretical guarantees that make myopic planning optimal for linear surrogates. This presents an interesting trade-off between model expressiveness and planning complexity that warrants further investigation. Techniques from approximate dynamic programming, such as lookahead policies and rollout algorithms (Bertsekas, 2024), could potentially bridge this gap by enabling limited-horizon planning with learned value function approximations.

## O.2. Theoretical Extensions

Several theoretical extensions could broaden the applicability of our framework:

**Weak Submodularity.** Extending our analysis beyond monotone submodularity to weaker notions like weak submodularity (Santiago & Yoshida, 2020) could better capture real-world utility functions that only approximately satisfy submodular properties. Many practical machine learning objectives exhibit diminishing returns behavior that is close to, but not exactly, submodular.

**Stochastic Selection.** Considering distributions over sequences rather than deterministic selections could lead to more robust selection strategies, particularly when utility estimates are uncertain. This connects to recent work on stochastic submodular optimization and could provide confidence intervals for selection quality.

**Beyond Curvature.** While curvature provides a useful characterization of approximation quality degradation, other structural properties of utility functions (such as noise stability or smoothness) might yield tighter or complementary bounds for specific application domains.

## O.3. Algorithmic Improvements

The practical implementation of our framework could be enhanced through several algorithmic improvements:

**Alternative Distance Metrics.** Investigating different similarity measures for bipartite graph construction beyond Euclidean distance could better capture domain-specific relationships between data points. For instance, learned embeddings, kernel-based similarities, or task-specific distance functions might improve coverage-utility alignment in specialized domains.

**Efficient Threshold Search.** Implementing more sophisticated optimization techniques like Bayesian optimization or gradient-free methods could improve the efficiency of threshold selection in our bipartite graph construction, particularly for high-dimensional feature spaces where the optimal threshold varies significantly.

**Preprocessing Strategies.** Developing techniques to handle redundant data points through clustering or dimensionality reduction could address cases where high curvature limits theoretical guarantees. Such preprocessing could reduce effective curvature while preserving the essential structure of the selection problem.

**Scalability.** For very large datasets, approximate nearest neighbor search and graph sparsification techniques could further improve computational efficiency while maintaining selection quality, enabling application to million-scale data pools.

