# OpenReview forum: "Unifying and Optimizing Data Values for Selection via Sequential Decision-Making"
_ICML.cc/2026/Conference — ICML 2026 spotlight_

### Official Review · Reviewer_p2eR · 2026-03-12

**Soundness:** 3
**Presentation:** 2
**Significance:** 2
**Originality:** 3
**Overall Recommendation:** 4
**Confidence:** 3

**Summary:**

The authors aim to analyze a central area in data selection for machine learning: how to choose training subsets that maximize downstream model utility. The paper frames data selection as a sequential decision problem and shows that several existing data valuation methods can be interpreted as myopic policies derived from surrogate utility functions. Building on this view, the authors propose a bipartite coverage surrogate that links training and validation points and can be optimized efficiently with greedy selection.

Overall, this manuscript's central contribution concerns providing a unified sequential formulation of data selection, theoretical analysis relating existing methods to this formulation, and a practical surrogate-based method that performs competitively in several empirical evaluations. The ideas are interesting and the empirical results are promising, but some of the theoretical claims appear stronger than what is fully supported by the practical formulation and experiments.

**Compliance With Llm Reviewing Policy:**

Affirmed.

**Final Justification:**

Thank you for the rebuttal! I'm maintaining my rating of weak accept: this is a strong paper but the concerns I listed under limitations in my original review still stand, and this is what makes me a bit less excited about this paper.

**Key Questions For Authors:**

N/A

**Limitations:**

No. The paper does not explicitly discuss several important limitations of the work. I recommend that the authors add a limitations section acknowledging the following points.  Discussing them explicitly would improve the clarity and credibility of the paper.

L1. Some of the theoretical guarantees appear quite loose and depend on assumptions that are unlikely to hold exactly in practice. The paper would benefit from clarifying that the bounds are worst-case guarantees and may not tightly characterize performance in realistic settings.

L2. There is a noticeable gap between the theoretical formulation and the practical algorithm used in experiments. The theory is derived for the true utility function, whereas the empirical method relies on a bipartite surrogate whose relationship to the original objective is only partially justified. Explicitly acknowledging this mismatch would strengthen the paper.

L3. The empirical evaluation is conducted primarily on relatively small datasets and simple models. While these experiments are useful for controlled comparison, they limit the conclusions that can be drawn about performance in larger-scale or more realistic settings.

L4. The practical method relies heavily on the quality of the embedding or similarity representation used to construct the bipartite graph. If the representation is poor, the surrogate may not approximate the true utility well. This dependency should be discussed as a potential limitation of the approach.

**Strengths And Weaknesses:**

The paper has several strengths.

S1. Interesting conceptual framing. The paper proposes a clear and appealing view of data selection as a sequential decision problem. This perspective helps organize and interpret several existing approaches to data valuation and subset selection within a common framework.

S2. Technically sound core formulation. The dynamic programming formulation of the ranking objective and the interpretation of existing data valuation methods as myopic policies under surrogate utilities appear technically reasonable and are presented clearly.

S3. Practical surrogate method. The proposed bipartite coverage surrogate provides a computationally efficient approximation to the utility objective and can be optimized with simple greedy procedures. This makes the method relatively easy to implement and potentially practical.

S4. Empirical evaluation across multiple datasets. The paper includes experiments on several datasets and compares against multiple baselines. The results suggest that the proposed approach performs competitively in the evaluated settings.

S5. Insight into relationships between existing methods. The reinterpretation of Shapley-style and related data valuation methods within the sequential decision framework is conceptually interesting and may help clarify connections between previously separate lines of work.

There are also several weaknesses.

W1. Gap between theory and practical method. A central concern is the mismatch between the theoretical formulation and the practical algorithm used in the experiments. The theoretical guarantees are derived for the true utility function, while the proposed bipartite surrogate is only loosely connected to that objective and relies on assumptions that are not empirically validated.

W2. Some theoretical claims appear somewhat overstated. The paper presents a strong narrative about unifying existing methods and providing guarantees, but the experiments do not clearly demonstrate that the theoretical insights translate into substantial practical advantages.

W3. Limited experimental scope. Although several datasets are included, many experiments involve relatively small training sets and simple models. This makes it difficult to fully assess how the approach would perform in larger-scale or more realistic settings.

W4. Lack of explicit discussion of limitations. The paper does not explicitly discuss several important limitations, including the reliance on embedding quality for the bipartite surrogate and the strength of the assumptions required for the theoretical guarantees.

W5. Novelty lies mainly in the reframing. The primary novelty is the conceptual framing of data selection as a sequential decision problem and the reinterpretation of existing methods within this view. The practical algorithm itself is closer to a surrogate-based greedy selection strategy than a fundamentally new method.

---

> ### Author Rebuttal · Authors · 2026-03-31
>
> **We thank reviewer p2eR for the detailed suggestions and recognition of the theoretical unification!**
>
> > W1. Theory-practice gap; sandwich conditions rarely hold.
>
> **A1.** We agree the paper should separate three layers more sharply. (1) **True-utility theory:** DP formulation, regression views (Thm. 4.1), curvature bounds (Thm. 4.4) hold for the true utility $U$ with no surrogate assumptions. (2) **Surrogate guarantee:** coverage $\hat{U}$ is monotone/submodular, so greedy on $\hat{U}$ has the standard $(1{-}1/e)$ guarantee. (3) **End-to-end transfer:** Thm. H.2 connects $\hat{U}$ to $U$ only under sufficient sandwich conditions. Remark H.3 already states these may fail in practice.
>
> Our empirical claim is narrower: $\hat{U}$ is a much better predictor of $U$ than linear baselines (Table 3: MSE 0.019 vs0.089), and this translates to better selection (Table 9: avg 0.75 vs 0.713 for the strongest prior baseline). We ran a calibration analysis on 300 held nout subsets per run across all 8 datasets (3 seeds):
>
> | Metric | Value |
> |---|---:|
> | Mean Spearman $\rho$ ($\hat{U}$ vs $U$) | 0.718 |
> | Pairwise order preservation | 70.7% |
> | Lower-bound hold rate ($\epsilon$=0.2) | 85.9% |
> | Full sandwich hold rate ($\epsilon$=0.2) | 30.8% |
>
> The surrogate carries strong ranking signal. This is what matters for selection. The full sandwich holds only ~31%, so Thm. H.2 should be treated as idealized. We will move this three-layer clarification to the main text.
>
> > W2. Loose curvature bounds.
>
> **A2.** We agree $(1{-}c)^2$ is a worst-case guarantee, not a tight bound. This is the intended reading of Remark G.5. Thm. 4.4 is most useful as a diagnostic result: when myopic methods fail, high curvature gives a principled explanation (validated in RQ2, Fig. 2). We will clarify this in the revised version.
>
> > W3. Small-scale experiments; only Logistic Regression.
>
> **A3.** The three scales are deliberate:
>
> | Setting | $n$ | Model | Purpose |
> |---|---|---|---|
> | RQ1 | 20 | Logistic Reg. | Oracle DP benchmark ($O(n \cdot 2^n)$) |
> | RQ3 | 50-100 | Logistic Reg. | Practical selection, 8 datasets, 20 runs |
> | DATE-LM | 200k | Llama 3.1-8B | LLM-scale transfer (MMLU/GSM8K/ BBH) |
>
> Small $n$ in RQ1 is a design choice, not a limitation: exact DP is only tractable at that scale. Logistic Regression follows standard data valuation practice [1, 2, 3]. We will make this "oracle vs. practical transfer" split more explicit.
>
> > W4. No Limitations section.
>
> **A4.** Thank you for the reminder! We will add a limitations sec. covering the key gaps such that sandwich conditions may not hold, embedding quality dependence, etc. We note that the embedding ablation in Tab. 8 already provides partial evidence on the last point, but an explicit discussion is also good to have.
>
> > W5. Novelty is just reframing?
>
> A5. We also agree that the sequential reformulation is our central conceptual contribution. We believe a good framing opens up angles and insights that were previously out of reach, and that is the case here. The reframing directly yields new technical results: an exact DP characterization of the ranking objective, regression and ADP interpretations of Banzhaf and Beta Shapley that did not exist before, and a curvature based explanation for the inconsistent performance patterns observed in[4]. On the practical side, it guided us to the bipartite surrogate, which achieves 4.7x lower MSE than linear baselines and the highest average selection accuracy across all eight datasets (Table 9: 0.75 vs 0.713 for the strongest prior method). These results would not have emerged without the sequential lens.
>
> **Ref.**
>
> [1] Ghorbani and Zou, Data Shapley: Equitable Valuation of Data for Machine Learning, ICML 2019.
>
> [2] Kwon and Zou, Beta Shapley: a Unified and Noise-reduced Data Valuation Framework, arXiv:2110.14049, 2021.
>
> [3] Wang and Jia, Data Banzhaf: A Robust Data Valuation Framework for Machine Learning, AISTATS 2023.
>
> [4] Wang et al., Rethinking Data Shapley for Data Selection Tasks: Misleads and Merits, ICML 2024 Oral.

---

> > ### Author Rebuttal · Reviewer_p2eR · 2026-04-01
> >
> > Thank you for the response.

---

### Official Review · Reviewer_sB49 · 2026-03-15

**Soundness:** 3
**Presentation:** 3
**Significance:** 3
**Originality:** 3
**Overall Recommendation:** 5
**Confidence:** 2

**Summary:**

This paper addresses a critical gap in data valuation and selection: while game-theoretic methods (e.g., Data Shapley, Banzhaf) are widely used for data selection, their theoretical foundations for this downstream task remain underexplored. The authors reformulate data selection as a finite-horizon sequential decision-making problem, deriving an exact dynamic programming (DP) solution for the optimal selection sequence. This framework unifies existing game-theoretic data valuation methods by revealing them as myopic linear approximations to the sequential problem, with performance degradation tied to the utility function’s curvature (via a (1−c) 2approximation bound). To bridge theoretical optimality and practical scalability, the authors propose a bipartite graph-based surrogate utility that preserves submodular structure, enabling efficient greedy selection with provable guarantees. Comprehensive experiments on 8 classical ML benchmarks and large-scale LLM fine-tuning (DATE-LM) demonstrate that the proposed method outperforms 9 state-of-the-art baselines, closing the performance gap between existing approximations and optimal DP selection.

**Compliance With Llm Reviewing Policy:**

Affirmed.

**Key Questions For Authors:**

1.How does the bipartite surrogate perform when utilities are non-submodular (e.g., complementary data samples)? Can you provide experiments on synthetic or real datasets with non-submodular utilities to validate the framework’s robustness?
2.For the threshold selection in Algorithm 1, how are candidate thresholds generated (e.g., percentiles, fixed grid) and how many candidates do you typically test? Could you provide ablation results on the impact of threshold selection strategy on performance?
3.Have you evaluated the proposed method on extreme-scale datasets (n/m > 1M) with high-dimensional features? If so, what optimizations (e.g., ANN, graph sparsification) did you use, and how did they impact selection quality?
4.How does your bipartite surrogate compare to recent specialized submodular data selection methods (e.g., coverage-based methods for LLM instruction tuning)? Do you observe unique advantages in terms of sample efficiency or computational cost?
5.Could you discuss potential biases in the bipartite selection framework? For example, if the validation set underrepresents certain groups, would the coverage objective amplify this bias, and how might this be mitigated?

**Limitations:**

1.Non-submodular utility analysis: The framework focuses primarily on submodular utility functions, which are common in data selection but do not cover all real-world scenarios (e.g., complementary data samples with non-diminishing marginal gains). The authors should analyze the bipartite surrogate’s performance under non-submodular utilities and quantify performance degradation when the utility violates submodularity assumptions.
2.Extreme-scale data scalability: While the method scales linearly with  n and  m, the paper does not evaluate performance on datasets where
n/m>1M(e.g., web-scale text datasets). The authors should address the computational bottlenecks of edge construction in the bipartite graph (O(
nmd)) and propose optimizations (e.g., approximate nearest neighbor, graph sparsification) to enable deployment at extreme scale.
3.Threshold selection reproducibility: Algorithm 1 relies on threshold search via distance distribution candidates, but the paper does not specify candidate generation strategies (e.g., quantiles, grid search) or the number of candidates tested. This ambiguity hinders reproducibility, and the authors should formalize threshold selection and provide ablation results on its impact.
4.Specialized submodular method comparison: The authors overlook recent submodular optimization methods tailored for data selection (e.g., coverage-based LLM instruction tuning methods). They should compare the bipartite surrogate to these specialized approaches to clarify its unique advantages in sample efficiency and scalability.
5.Ethical bias and societal impact: The impact statement is generic and does not address potential biases in the coverage-based selection objective. If the validation set underrepresents marginalized groups, the coverage objective may amplify underrepresentation. The authors should discuss this risk and propose mitigation strategies (e.g., group-aware validation sampling).

**Strengths And Weaknesses:**

strength：
1.Novel Theoretical Unification: The paper’s core contribution—framing data selection as a sequential decision-making problem—provides a transformative lens for understanding existing data valuation methods. By showing that Data Shapley, Banzhaf, and Beta Shapley are myopic linear approximations in approximate dynamic programming (ADP), the work resolves longstanding ambiguities about why these methods succeed or fail. The curvature-based analysis ((1−c) 2 bound) further clarifies performance limitations under submodularity, offering actionable insights for method selection across datasets.
2.Rigorous Theoretical Foundations: The paper delivers rigorous theoretical results, including: (1) regression characterizations of game-theoretic values (Shapley as constrained WLS, Banzhaf as unconstrained OLS), (2) optimality proofs for linear utilities, (3) curvature-dependent approximation guarantees for submodular utilities, and (4) end-to-end guarantees for the bipartite surrogate. These results are complemented by detailed proofs in appendices, ensuring theoretical soundness and reproducibility.
3.Practical and Scalable Solution: The proposed bipartite graph-based surrogate strikes an ideal balance between theoretical guarantees and computational feasibility. By leveraging submodular coverage utility, the method enables greedy selection with a (1−1/e)approximation ratio, while avoiding the exponential complexity of exact DP. The approach scales linearly with the number of training-validation pairs (nm) and is parallelizable, making it applicable to large-scale datasets (e.g., 200k instruction pools for LLM fine-tuning).
4.Comprehensive and Convincing Experiments: The empirical evaluation is exemplary, covering: (1) classical ML benchmarks (8 datasets) with 20 independent runs, (2) controlled experiments on utility curvature (via feature aggregation), (3) large-scale LLM fine-tuning (DATE-LM benchmark), (4) ablation studies on embedding backends and hyperparameters, and (5) comparisons to optimal DP selection (to quantify baseline gaps). Results consistently show the proposed method outperforms baselines by 1–10% across tasks, with particularly strong gains in sample-efficient selection (early-stage selection curves).
5.Broad Applicability: The framework is not limited to classical ML but extends seamlessly to LLM fine-tuning data selection, demonstrating its versatility across modalities (tabular, text, instruction data). The bipartite surrogate’s flexibility (compatible with diverse distance metrics/embeddings) further enhances its practical value for real-world data-centric ML pipelines.

Weaknesses
1.Limited Discussion of Non-Submodular Utilities: While the paper focuses on submodular utilities (a common case for data selection), it does not address how the framework performs when utilities violate submodularity. Real-world utilities may exhibit non-diminishing returns (e.g., complementary data samples), and the authors provide no analysis of the bipartite surrogate’s behavior in such scenarios.
2.Ambiguity in Threshold Selection for Bipartite Graphs: Algorithm 1 describes threshold search via candidate values from the distance distribution, but the paper does not clarify how candidate thresholds are generated (e.g., quantiles, grid search) or how many candidates are tested. This lack of detail could hinder reproducibility for practitioners.
3.Insufficient Analysis of Computational Scalability for Extreme-Scale Data: While the method scales linearly with nm, the paper does not evaluate performance on datasets where n or m exceed 1M (e.g., web-scale instruction datasets). It remains unclear how the bipartite graph’s edge construction (O(nmd) complexity) would scale to high-dimensional, extreme-scale data without approximate nearest-neighbor (ANN) optimizations.
4.Lack of Comparison to Recent Submodular Data Selection Methods: The paper compares to game-theoretic and learning-based baselines but overlooks recent submodular optimization methods specifically designed for data selection (e.g., coverage-based methods for LLM instruction tuning). This omission leaves unanswered whether the proposed bipartite surrogate offers unique advantages over specialized submodular approaches.
5.Superficial Discussion of Societal Impacts: The impact statement is generic and does not address potential biases in data selection—e.g., whether the bipartite surrogate’s coverage objective could amplify underrepresentation of marginalized groups in the validation set, leading to biased selection.

---

> ### Author Rebuttal · Authors · 2026-03-31
>
> **We thank Reviewer sB49 for the valuable suggestions and thoughtful questions!**
>
> > W1. Non-submodular utilities.
>
> **A1.** The sequential DP formulation (Prob. 3.1), DMDP reformulation (Sec. 3.1), regression views (Thm. 4.1), and ADP interpretation (Thm. 4.2) do **not** need submodularity. It enters only in the approximation analysis: Thm. 4.4 (curvature bound) and Prop. 5.2 (greedy guarantee). So the framework is broader, but the curvature based guarantees are scoped to normalized monotone submodular utilities; Thm. H.2 is separate and depends on the surrogate's submodularity and the sandwich conditions.
>
> Empirically, submodular behavior dominates data selection: Our Fig. 3 shows concave selection curves across all datasets. LESS [1] found that 5% of instruction data can outperform the full dataset in LLM fine-tuning. Complementary interactions exist but are not the dominant pattern.
>
> We agree this scope should be stated more prominently. Thm. 4.4 is specific to monotone submodular $U$. Thm. H.2 is a separate surrogate- ransfer result that depends on the sandwich conditions rather than submodularity of $U$. We will add this to a Limitations section.
>
> > W2. Threshold selection details.
>
> **A2.** We totally agree Algorithm.1 needs more detail and will make Sec.5.2 more precise in the revision. Our implementation works as follows: (1) compute train-valid similarities, keep only same-label pairs; (2) search candidate thresholds as quantiles of this intra-class similarity distribution (range [0.50,  0.95]); (3) for each candidate, sample random subsets (sizes from uniform ratio in [0.01, 0.99]), retrain, record validation accuracy; (4) pick the threshold that **minimizes average MSE** between coverage and validation accuracy. The constants in App.I are complexity-level descriptions; the implementation uses a denser grid. For reproducibility, see `scripts/run_threshold_sensitivity.py` and `rebuttal/threshold_sensitivity_summary.csv` in the anonymous repository. We ran an ablation across three grid densities (3 seeds, 2 datasets):
>
> | $N_\tau$ | nomao@20% | nomao@50% | digits@20% | digits@50% |
> |---:|---:|---:|---:|---:|
> | 30 | 0.877 | 0.884 | 0.802 | 0.851 |
> | 100 | 0.885 | 0.896 | 0.792 | 0.850 |
> | 1000 | 0.885 | 0.883 | 0.791 | 0.850 |
>
> Accuracy varies by about 1 percentage point. The result is not highly sensitive to grid density.
>
> > W3. Scalability beyond 200k?
>
> **A3.** DATE-LM uses a **200k** pool, selects 10k examples, and BipCov uses a top-L sparse graph that bypasses threshold search. Selection takes <1 second on CPU. We agree the paper shows 200k scale, not >1M. We will position ANN/graph sparsification as future work.
>
> > W4. Missing comparison with submodular baselines for LLM data selection.
>
> **A4.** Yes, this is indeed a fair point. Our DATE-LM baselines are per-example scoring methods; none explicitly optimizes a submodular objective. We are aware of several such methods. SMART [2] uses facility location to score tasks and allocate budget across task mixtures. MIG[3] builds a label graph and greedily selects samples that maximize information gain via a submodular objective. DiverseEvol [4] iteratively applies K center selection in the model's embedding space. These methods start from a submodular objective chosen a priori.
>
> Our approach takes a different path: we start from a sequential decision-making formulation (Prob. 3.1), identify that a faithful surrogate is the bottleneck (Thms. 4.2,4.3), and arrive at submodular coverage as the surrogate that best fits this role. This provides additional theoretical structure that the direct submodular approach does not address. We will add these works to related work.
>
> We also welcome the reviewer's suggestions on specific methods to include.
>
> > W5. Bias from validation set.
>
> **A5.** It definitely could be the case! Coverage is defined w.r.t. the validation set. If it underrepresents a subgroup, the selector may under prioritize relevant training data. We will expand the Impact Statement with mitigations: group-aware sampling etc.
>
> **Ref.**
>
> [1] Xia et al., LESS: Selecting Influential Data for Targeted Instruction Tuning, ICML 2024.
>
> [2] Renduchintala et al., SMART: Submodular Data Mixture Strategy for Instruction Tuning, ACL 2024.
>
> [3] Chen et al., MIG: Automatic Data Selection by Maximizing Information Gain in Semantic Space, ACL 2025.
>
> [4] Wu et al., Self-Evolved Diverse Data Sampling for Efficient Instruction Tuning, arXiv:2311.08182, 2023.

---

### Official Review · Reviewer_KAht · 2026-03-29

**Soundness:** 3
**Presentation:** 3
**Significance:** 3
**Originality:** 3
**Overall Recommendation:** 4
**Confidence:** 4

**Summary:**

The paper focuses on the problem of data selection/valuation, and analyzes that popular game-theoretic methods like Data Shapley are not explicitly optimized for choosing the best data-subsets across varying or evolving budget constraints. To solve this problem, the paper reformulates data selection as a finite-horizon sequential decision-making problem. And the optimal selection sequence of data samples can be derived through DP. Further, the paper analyzes game-theoretic valuations (Shapley, Banzhaf and Beta-Shapley) through the above problem formulation and conclude that they actually act as myoptic approximate DP policies and their optimality degrade quadratically with the curvature of the utility function.

**Compliance With Llm Reviewing Policy:**

Affirmed.

**Key Questions For Authors:**

1. In Algorithm 1, the optimal threshold $\tau^*$ is chosen by minimizing prediction error on random subsets. How sensitive is the downstream selection performance to the choice of $\tau^*$, and what is the practical computational cost of this search step on larger datasets?
2. Theorem H.2 relies on the strict condition $\hat{U}(S) \le U(S)$. Because coverage utility can easily overestimate actual model accuracy, how often do these conditions actually hold empirically on your tested datasets, and how does failure of this condition correlate with performance drops?

**Limitations:**

Yes

**Strengths And Weaknesses:**

Strengths:
- The formulation of data selection as a finite-horizon sequential decision-making problem is interesting. The limitation analysis of game-theoretic valuation methods makes sense and mathematically well-supported.
- The paper is well-written. It illustrates the problem formulation clearly, interprets ADP analysis and concludes with a practical bipartite solution.
- Reconciling data valuation with sequential decision-making is a good contribution to data-centric AI. The application of the bipartitie graph method to LLM instruction fine-tuning proves that the theoretical framework scales to modern problems.

Weaknesses:
- The paper formulates data selection as a sequential process with strict nested constraints ($S_{k-1}^\pi \subseteq S_k^\pi$), However, the practical deployment scenario for this specific constraint is not clearly justified. In many real-world machine learning pipelines, data subsets of size $k$ are constructed independently to maximize utility (or at least only parts of existing data samples should be reserved and others can be  discarded and replaced to compose a more optimal subset). Unless the authors explicitly assume a high marginal cost for swapping/discarding data points (e.g., expensive iterative human labeling or data accumulation), enforcing a strict sequential sequence artificially limits the selection space.
- A core premise of the paper is that optimal data valuation requires considering both immediate utility rewards and future value through a Dynamic Programming (DP) lens. However, the proposed practical solution (the bipartite graph surrogate) ultimately reverts to a myopic, greedy selection policy. By ignoring future value in the practical implementation , the proposed algorithm acts as another approximate dynamic programming (ADP) heuristic, suffering from the same myopic limitations as the existing game-theoretic baselines it critiques

---

> ### Author Rebuttal · Authors · 2026-03-30
>
> **We thank Reviewer KAht for the suggestions and careful reading of the theoretical claims!**
>
> > W1. Nested constraint is restrictive; why not fixed-k optimization?
>
> **A1.** Our sequential objective formalizes the **standard ranking-based data selection protocol**. A single ordering is evaluated across budgets through progressively larger prefixes (Def. 2.5; [1 (Supp Fig. 1b), 2 (Fig. 11)]). The nested structure is not an extra assumption. It is the mathematical structure of the objective itself: Prob. 3.1 maximizes average utility across all prefix sizes.
>
> This is practically motivated in two ways. First, practitioners often do not know the budget $k$ in advance. A single ranking across all sizes is more useful than separate subsets each needing its own optimization. Second, data is continuously generated in practice. New samples arrive over time, budgets expand, and earlier acquisitions are retained. Data acquisition is therefore an ongoing process, not a one-shot event. A ranking that supports stopping at any point with the best subset up to that size directly fits this reality.
>
> Admittedly, if one targets a single fixed budget with unrestricted replacement, the natural problem is fixed $k$ optimization, not Prob. 3.1. We will clarify this scope distinction.
>
> > W2. Bipartite method is also myopic; contradicts the critique of existing methods.
>
> **A2.** Yes, this is a great observation! This is intentional, not contradictory. Our framework separates suboptimality into two sources (Sec. 3.1, Fig. 4): **(1) surrogate quality** and **(2) policy myopia**. For a linear surrogate, marginal gains are state-independent (Thm. 4.2); under linear utility, myopic ranking is optimal (Thm. 4.3). The limitation of existing methods is not greedy selection itself but the linear surrogate's poor fit to the true utility. Our contribution is replacing the linear surrogate with a more faithful bipartite/submodular one within the same tractable greedy regime. Tab. 3 validates this (MSE 0.019 vs 0.089). Tab. 9 shows better selection than the strongest prior baseline (avg 0.75 vs 0.713). Non-myopic planning is discussed in Sec. 6 and App. P.1 as future work.
>
> > Q1. Threshold search cost and sensitivity?
>
> **A3.** In the classical setting, cost is dominated by subset retraining, not graph construction. We ran a sensitivity ablation across three grid densities (3 seeds, 2 datasets):
>
> | $N_\tau$ | nomao@20% | nomao@50% | digits@20% | digits@50% |
> |---:|---:|---:|---:|---:|
> | 30 | 0.877 | 0.884 | 0.802 | 0.851 |
> | 100 | 0.885 | 0.896 | 0.792 | 0.850 |
> | 1000 | 0.885 | 0.883 | 0.791 | 0.850 |
>
> Accuracy varies by about 1 percentage point across settings. The dense classical search ($N_\tau$=1000) takes ~11s on nomao, ~16s on digits, and ~37s on bbc-embeddings (the slowest). It remains a seconds scale step. On DATE-LM (200k pool), BipCov uses a top-L sparse graph (L=200) that bypasses threshold search. Selection takes <1 second on CPU (Tab. 5).
>
> > Q2. Sandwich conditions: how often do they hold? Correlation with performance?
>
> **A4.** We do **not** claim the sandwich conditions generally hold. We ran a calibration analysis (300 held-out subsets x 3 seeds x 8 datasets; see `scripts/run_h2_calibration.py` and `rebuttal/` in the anonymous repository):
>
> | Dataset | Spearman | Lower @0.2 | Sandwich @0.2 |
> |---|---:|---:|---:|
> | 2dplanes | 0.728 | 82.9% | 13.2% |
> | nomao | 0.607 | 94.9% | 43.8% |
> | bbc-embed | 0.859 | 93.9% | 43.4% |
> | MiniBooNE | 0.619 | 84.8% | 17.7% |
> | digits | 0.907 | 88.7% | 61.4% |
> | election | 0.655 | 83.0% | 31.8% |
> | electricity | 0.662 | 75.8% | 20.3% |
> | fried | 0.704 | 83.6% | 15.1% |
> | Mean | 0.718 | 85.9% | 30.8% |
>
> The full sandwich holds only ~31%. Thm. H.2 is an idealized sufficient condition. But the surrogate preserves strong ranking signal (mean Spearman 0.718). This is what drives practical selection quality. To measure downstream impact, we define drop@b% = full dataset test accuracy minus accuracy of the selected top b% subset. Moderate-budget performance stays strong (mean drop@ 50% = 0.02) even though sandwich satisfaction is infrequent. Early budget performance is more variable (mean drop@ 20% = 0.095), with election being the clearest hard case. Exploratory run level associations between sandwich satisfaction and performance are weak and inconsistent across budgets and datasets, so we do not claim a clean monotonic relation. We will clearly separate the transfer theorem from the empirical claim.
>
> **Ref**
>
> [1] Ghorbani and Zou, Data Shapley: Equitable Valuation of Data for Machine Learning, ICML 2019.
>
> [2] Kwon and Zou, Beta Shapley: a Unified and Noise-reduced Data Valuation Framework, arXiv:2110.14049, 2021.

---

### Decision · Program_Chairs · 2026-04-30

**Decision:**

Accept (spotlight)

**Comment:**

This is an interesting papers that provides a unifying view on existing game-theoretic data valuation methods. The authors provided convincing rebuttal answers and the reviewers broadly support acceptance.